# Rethinking Object Detection and Tracking

Arpit Sharma
Visvesvaraya Technological
University
Bengaluru, Karnataka, India
arpit.sharma@mtech.christuniversity.in

Lei Wang*
Griffith University
Brisbane, Queensland, Australia
Data61/CSIRO
Canberra, Australian Capital
Territory, Australia
l.wang4@griffith.edu.au

Yongsheng Gao*
Griffith University
Brisbane, Queensland, Australia
yongsheng.gao@griffith.edu.au

## Abstract

Recent years have witnessed a profound transformation in object detection and tracking, driven by advances in transformers, diffusion models, multimodal learning, and large-scale pretraining. Beyond performance gains, the field is undergoing a conceptual shift, from closed-set, task-isolated pipelines toward open-world, multi-task, and semantically grounded visual perception systems. This survey provides a review of very recent object detection and tracking research, systematically analyzing more than one hundred representative works across 2D, 3D, multi-view, multimodal, and vision-language settings. By consolidating models, datasets, evaluation protocols, and targeted challenges, we expose cross-task patterns that are often overlooked in existing surveys. Our analysis shows several emerging trends: the convergence of detection and tracking into unified formulations, the growing role of generative and diffusion-based temporal modeling, the rise of open-vocabulary and language-conditioned tracking, and the increasing importance of uncertainty modeling and multimodal fusion in 3D and adverse environments. In addition, we provide a quantitative analysis of dataset usage, evaluation metrics, and challenge prevalence over time, highlighting how benchmark choices and metric design shape research directions. The survey concludes by identifying open problems and underexplored intersections, such as scalable open-world tracking, unified evaluation across modalities, and principled handling of uncertainty and semantics, that point toward the next phase of visual perception research. By offering both breadth and synthesis, this work aims to serve as a reference and a roadmap for future advances in object detection and tracking. Our appendix is available here.

## CCS Concepts

• **General and reference** → **Surveys and overviews**; **General literature**; • **Computing methodologies** → *Object detection*; *Tracking*; Machine learning.

## Keywords

Object Detection, Object Tracking, Multi-Object Tracking, Joint Detection and Tracking, Temporal Modeling, Open-World Perception

*Corresponding authors.

*WWW Companion '26, Dubai, United Arab Emirates*
© 2026 Copyright held by the owner/author(s).
ACM ISBN 979-8-4007-2308-7/2026/04
https://doi.org/10.1145/3774905.3794677

**ACM Reference Format:**
Arpit Sharma, Lei Wang, and Yongsheng Gao. 2026. Rethinking Object Detection and Tracking. In *Companion Proceedings of the ACM Web Conference 2026 (WWW Companion '26), April 13–17, 2026, Dubai, United Arab Emirates.* ACM, New York, NY, USA, 20 pages. https://doi.org/10.1145/3774905.3794677

## 1 Introduction

Object detection and tracking are foundational problems in computer vision [15, 36–38], enabling machines to localize, recognize, and follow objects across space and time. Over the past decade, progress in these areas has been largely driven by advances in deep learning architectures and large-scale benchmarks, resulting in remarkable performance improvements under controlled settings [22–25, 32]. However, as visual perception systems are increasingly deployed in unconstrained real-world environments, ranging from autonomous driving and aerial surveillance to sports analytics, robotics, and medical imaging, the limitations of classical problem formulations have become increasingly apparent [2, 3, 6, 7, 14].

Recently, the field has entered a period of rapid diversification and conceptual transition. Object detection has expanded beyond closed-set recognition toward open-vocabulary, open-domain, and weakly supervised settings, while tracking has evolved from short-term, appearance-driven association to long-term, semantics-aware, and multimodal temporal reasoning [19, 30]. At the same time, the boundaries between detection and tracking are becoming increasingly blurred: joint detection-tracking frameworks, end-to-end identity prediction, diffusion-based temporal interpolation, and unified multi-task models are replacing traditional modular pipelines [4, 51, 54, 55]. These developments signal a shift in emphasis, from isolated task optimization toward robust, generalizable, and semantically grounded visual perception.

Despite this rapid progress, the literature has become highly fragmented [9, 17, 46, 47]. Recent methods differ not only in architectures (*e.g.*, transformers, diffusion models, state-space models, and vision-language systems) but also in assumptions about supervision (*e.g.*, modality, task definition, and evaluation protocol) [1, 28]. As a result, comparing approaches across tasks and domains has become increasingly difficult. Moreover, many surveys continue to organize the field along traditional task boundaries (*e.g.*, detection *vs.* tracking, 2D *vs.* 3D), which obscures deeper connections between methods that are designed to address the same underlying challenges, such as occlusion, long-term identity preservation, fast and non-linear motion, small or low-visibility objects, domain and category shift, and real-time constraints [39, 48].

This survey is motivated by the observation that challenges (not task labels), now define the research frontier of object detection and

tracking. Across diverse settings, recent works repeatedly target a shared set of bottlenecks that limit real-world deployment. Occlusion and identity confusion dominate multi-object tracking and sports analytics; fast motion and sparse observations challenge UAV, event-based, and satellite video analysis; small-object detection and low illumination appear across aerial, underwater, and infrared domains; and domain shift, data scarcity, and open-set recognition arise in open-vocabulary, few-shot, and multimodal perception. By examining how these challenges are addressed across tasks, modalities, and architectures, we can extract insights that are not visible through task-centric reviews.

To this end, we conduct a review of object detection and tracking research published recently (within the past three years) in leading computer vision and machine learning venues. Each selected work is analyzed along consistent dimensions, model category, datasets, evaluation metrics, task formulation, and targeted challenges, allowing systematic comparison across otherwise disparate research threads. Beyond qualitative discussion, this survey provides a quantitative analysis of the field. By aggregating dataset usage, evaluation metrics, and challenge prevalence over time, we show how benchmark choices and evaluation practices shape research priorities. Our analysis highlights the dominance of standard multi-object tracking and detection benchmarks, alongside a growing reliance on 3D, multi-view, RGB-D, thermal, event-based, and vision-language datasets. Similarly, while traditional metrics, *e.g.*, mean average precision (mAP), higher order tracking accuracy (HOTA), *etc.*, remain central, newer evaluation protocols increasingly emphasize long-term consistency, semantic correctness, and robustness under adverse conditions. These trends reflect a broader shift toward holistic and deployment-oriented evaluation.

Ultimately, this survey aims not only to summarize recent advances but also to synthesize the field's trajectory. By connecting architectural trends with the challenges they aim to solve, we identify emerging unifying principles, such as temporal generative modeling, semantic grounding via language, uncertainty-aware perception, and multimodal fusion, that cut across detection and tracking paradigms. We also highlight open problems and underexplored intersections that are likely to shape the next phase of research. The main contributions of this survey are threefold:

i. A challenge-driven synthesis of object detection and tracking research. We reorganize the literature around fundamental real-world challenges rather than traditional task boundaries, showing shared solutions and transferable insights across detection, tracking, 3D perception, and multimodal vision.

ii. A structured, large-scale comparative analysis of modern perception systems. Through unified and quantitative trend analysis, we systematically compare models, datasets, metrics, and problem settings, exposing patterns, gaps, and biases in current benchmarks and evaluation practices.

iii. A forward-looking perspective on emerging paradigms and open problems. We identify key architectural and conceptual shifts, such as unified detection-tracking models, diffusion-based temporal reasoning, vision-language grounding, and uncertainty-aware modeling, and outline future research directions necessary for scalable, open-world visual perception.

## 2 Our Taxonomy of Detection and Tracking

Object detection and tracking have historically evolved as distinct research problems, each supported by separate benchmarks, architectures, and evaluation criteria. Detection has primarily focused on spatial localization and category recognition within individual frames, while tracking emphasizes temporal association and identity preservation across time. Recent progress, however, increasingly blurs this separation. Shared real-world challenges, including occlusion, fast or irregular motion, domain shift, long-term temporal dependencies, and multimodal uncertainty, have driven the development of methods that span, combine, or even redefine traditional task boundaries. To systematically capture this evolution, we organize contemporary detection and tracking methods along four complementary axes: task formulation, supervision and openness, spatial-temporal scope, and sensing modality.

**Task formulation: from modular pipelines to unified temporal reasoning.** The dominant paradigms in detection and tracking can be distinguished by how they formulate spatial localization and temporal association. Classical object detection largely follows a closed-set formulation, where models are trained to recognize a fixed set of categories [37]. Despite its maturity, this setting remains central to many practical systems and continues to serve as the backbone for tracking-by-detection pipelines [2]. However, recent years have seen a clear shift toward more flexible formulations. Open-vocabulary and open-set detectors extend recognition beyond predefined categories by using semantic embeddings or vision-language alignment [12, 33, 49]. While these approaches significantly improve category generalization, they also introduce semantic ambiguity and confidence uncertainty, which directly impact downstream tracking stability.

In tracking, the traditional tracking-by-detection paradigm decouples detection from association, offering modularity and scalability but remaining sensitive to detection noise, missed objects, and identity switches under occlusion [2]. To address these limitations, joint detection-tracking frameworks integrate localization and association within a single optimization objective, enabling tighter temporal coupling at the cost of increased architectural complexity [50]. More recently, a growing body of work reframes tracking as a direct prediction problem, where object trajectories or identities are inferred as sequences or sets without explicit association steps. Transformer-based and diffusion-based trackers exemplify this trend, offering improved long-term consistency and robustness to intermittent observations [27, 44]. At the same time, the scope of tracking itself has expanded beyond classical multi-object tracking to include long-term single-object tracking [20, 42], point-level tracking [5, 11], and generic visual correspondence [41, 45], further dissolving rigid task boundaries.

**Supervision and openness: scaling beyond exhaustive annotations.** Another defining axis of modern detection and tracking research concerns the degree of supervision and the assumptions made about the operating environment. Fully supervised methods, trained with dense spatial and temporal annotations, continue to dominate benchmark performance and remain the standard in controlled settings [51]. However, their reliance on extensive labeling limits scalability, especially for long videos, rare categories, and new sensing modalities.

**Table 1: High-level summary of dominant trends in recent object detection and tracking literature (2024-2025).**

| Aspect | Dominant Category | Representative Datasets | Common Metrics | Task Scope | Key Insight |
|---|---|---|---|---|---|
| Task formulation | Multi-object tracking; unified detection-tracking | MOT17, MOT20, Dance-Track, SportsMOT | HOTA, IDF1, MOTA | Multi-object tracking; joint detection and tracking | Shift from modular pipelines toward unified temporal reasoning |
| Model paradigm | End-to-end; diffusion-based; transformer-based | Standard MOT and SOT benchmarks | HOTA, IDF1, AUC | Tracking | Growing adoption of end-to-end and generative modeling paradigms |
| Sensing modality | RGB-dominant with increasing multimodality | RGB-D, LiDAR, event-based, vision-language datasets | mAP, HOTA, Precision | Detection, Tracking | Multimodal fusion increasingly used for robustness |
| Spatial reasoning | 3D-aware and BEV-based reasoning | KITTI, nuScenes, Waymo, multi-view datasets | 3D mAP, HOTA, NDS | 3D detection, 3D multi-object tracking | Persistent 3D perception is becoming a central design goal |
| Supervision regime | Weakly supervised; open-world; self-supervised | Open-vocabulary and domain-generalized benchmarks | mAP, Recall | Open-world detection, open-world tracking | Reduced reliance on dense annotations and closed-set assumptions |
| Datasets | Legacy benchmarks + domain-specific datasets | MOT-style, UAV, sports, medical, multimodal | HOTA, IDF1, mAP | Cross-domain | Diversification beyond generic benchmarks is accelerating |
| Evaluation metrics | Joint detection-association metrics | MOT-style benchmarks | HOTA, IDF1 | Tracking | Emphasis on identity consistency over frame-wise accuracy |
| Challenges | Occlusion, identity switches, long-term association | Crowded, long-horizon datasets | HOTA, IDF1 | Multi-object tracking, vision-language tracking | Temporal consistency remains the dominant bottleneck |

In response, weakly supervised, self-supervised, and unsupervised approaches have gained increasing attention [10, 11, 53]. These methods exploit temporal consistency, motion cues, cross-modal alignment, or proxy objectives to reduce annotation cost while maintaining competitive performance. Such strategies are particularly important for large-scale or domain-specific scenarios where annotations are sparse or expensive. Closely related is the transition from closed-world assumptions to open-world settings, where models need to handle unseen categories, evolving scenes, and ambiguous supervision over time [21, 34]. Open-world detection and tracking expose a fundamental tension between semantic flexibility and identity consistency: while models become more adaptable, maintaining stable object identities under semantic uncertainty remains a major challenge. This tension is a recurring theme across many of the methods summarized in our tables.

**Spatial and temporal scope: from 2D frames to 3D perception.** Detection and tracking methods can be distinguished by the spatial and temporal scope of their representations. The majority of existing work operates in 2D image space, benefiting from abundant data and mature architectures [18, 51]. However, 2D representations struggle with severe occlusion, depth ambiguity, and viewpoint changes. As a result, there is growing interest in 3D perception, where geometry from monocular cues, LiDAR, or multi-view reconstruction enables more physically grounded reasoning about object motion and interaction [16, 26, 40]. Temporal scope further differentiates short-term tracking (which focuses on local frame-to-frame continuity) from long-term and persistent tracking (which requires memory, re-identification), and recovery after prolonged absence or occlusion [8, 13]. Methods targeting longer temporal horizons increasingly incorporate explicit memory mechanisms, global context modeling, or trajectory-level reasoning [13, 16]. Notably, many recent approaches operate across both expanded spatial and temporal scopes, e.g., 3D multi-object tracking over long sequences [16, 51], highlighting a trend toward persistent scene understanding rather than transient frame-level predictions.

**Sensing modality: from single sensors to multimodal perception.** Detection and tracking paradigms vary significantly in

their use of sensing modalities. Single-modality systems, particularly RGB-based models, remain the most common due to simplicity and data availability [31]. However, they are inherently vulnerable to adverse conditions, e.g., low illumination, motion blur, and occlusion. To mitigate these limitations, an increasing number of methods integrate complementary modalities, including depth, LiDAR, thermal imaging, event streams, audio, and language [26, 56].

Multimodal systems offer improved robustness and richer semantic cues but introduce new challenges related to modality alignment, fusion strategy, and missing or noisy signals [35, 52]. Recent work explores both early and late fusion, adaptive weighting, and modality-agnostic architectures, reflecting the lack of consensus on a universally optimal solution [43]. Importantly, multimodality often interacts with other axes in this taxonomy: for instance, open-vocabulary tracking frequently relies on language [21], while long-term tracking benefits from geometry-aware or event-based sensing [29]. These interactions underscore the need for holistic design choices rather than isolated architectural decisions.

A glossary of commonly used evaluation metrics, including their full names, focus, and typical use cases, is provided in Appendix.

## 3 Quantitative Trend Analysis

**Analysis protocol and scope.** We surveyed peer-reviewed publications from major computer vision and machine learning venues, including CVPR, ICCV, ECCV, NeurIPS, ICLR, TPAMI, and IJCV. Each method was annotated along multiple axes: task formulation (e.g., detection, single-object tracking, multi-object tracking, joint detection-tracking, 3D tracking), sensing modality (e.g., RGB, LiDAR, RGB-D, thermal, event-based, vision-language), architectural paradigm (e.g., CNN-based, transformer-based, diffusion-based, state-space or recurrent), supervision level, and the primary challenges explicitly targeted. Methods were allowed to belong to multiple categories to reflect multi-task and multimodal designs. Aggregate counts and proportions were computed and visualized in Figures 1, 2, and 3 . This protocol enables a holistic view of how research priorities have evolved over time.

To ensure methodological transparency, we explicitly specify the inclusion and categorization criteria underlying our quantitative

analysis. We consider peer-reviewed conference and journal papers (published between 2024 and 2025) that introduce, benchmark, or conduct substantive analyses of object detection, tracking, or their unified formulations. Survey articles, purely application-driven studies lacking methodological contributions, and extended arXiv versions of previously published works are excluded to avoid redundancy. As many contemporary approaches span multiple paradigms, we use a multi-label annotation scheme rather than enforcing mutually exclusive categories. Accordingly, a single work may contribute to multiple trend counts, reflecting the inherently overlapping nature of modern research directions. This design choice enables a more faithful characterization of the field's structural diversity, rather than imposing artificially disjoint taxonomies.

To construct the quantitative trends, we manually extracted information on challenges, datasets, and evaluation metrics from each selected paper. Specifically, challenges were identified from the problem formulation, introduction, and discussion sections, while datasets and metrics were obtained from the experimental and evaluation sections. When a single paper addressed multiple challenges, datasets, or metrics, all relevant entries were recorded using a multi-label annotation scheme. This manual curation was performed to ensure semantic consistency and to avoid errors arising from automated keyword-based extraction, particularly for implicit or cross-cutting concepts. This procedure enables a faithful representation of how contemporary works position their contributions and report empirical performance.

Although this survey conceptually spans literature published between 2022 and 2025, the quantitative trend analysis in this section is intentionally restricted to papers from 2024 to 2025. This design choice reflects our aim to characterize the most recent paradigm shifts, emerging problem formulations, and evolving evaluation practices, as captured by the large comparative tables curated for this study. Earlier works from 2022-2023 are extensively incorporated into the related work, taxonomy, and qualitative synthesis, where they play a foundational role in shaping the conceptual structure of these trends. By separating long-term conceptual grounding from short-term quantitative dynamics, we aim to provide both historical context and a focused view of the field's current trajectory.

To complement the qualitative review, we conduct a quantitative analysis of recent object detection and tracking literature. By aggregating statistics across the methods summarized in Appendix, this analysis shows empirical trends in task focus, architectural paradigms, sensing modalities, datasets, evaluation metrics, and the challenges most frequently addressed by recent work. Rather than reporting raw counts in isolation, we interpret these trends to expose structural shifts in how the field defines problems and measures progress. To improve readability and provide a compact overview of the main empirical patterns, we summarize the dominant trends observed in Table 1. This summary complements the detailed method-level tables in Appendix by highlighting the most salient structural shifts in the field.

**Task formulation trends and convergence.** Object detection and multi-object tracking remain the most prevalent research topics, together accounting for the majority of recent publications. However, a closer examination reveals a steady rise in works addressing joint detection-tracking, long-term tracking, and point-level or

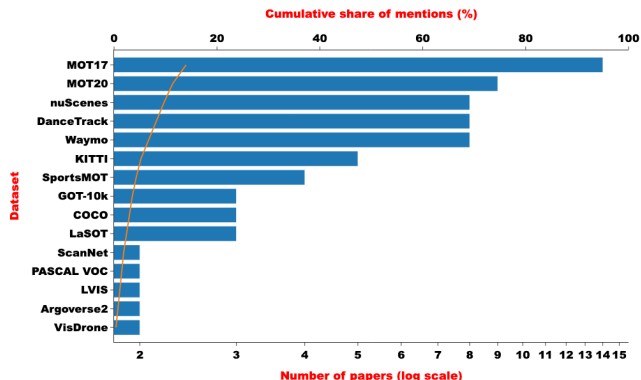

**Figure 1: Dataset usage across recent object detection and tracking literature (2024-2025). Horizontal bars indicate the raw frequency of benchmark dataset mentions extracted from the Dataset column of the literature review in Appendix, plotted on a logarithmic bottom $x$-axis to accommodate the wide dynamic range of usage counts. The overlaid curve, shown on the top $x$-axis, represents the cumulative share of mentions. For clarity, only the top-15 most frequently cited datasets are shown along the $y$-axis. The distribution shows the dominance of established multi-object tracking benchmarks (*e.g.*, MOT17, MOT20, DanceTrack, *etc.*) and illustrates the long-tailed nature of dataset adoption across diverse tasks and sensing modalities.**

dense tracking formulations. These trends indicate a gradual erosion of rigid task boundaries in favor of unified formulations that jointly reason about localization, identity, and temporal continuity.

This shift reflects growing recognition that frame-local optimization is insufficient for real-world scenarios involving prolonged occlusion, complex interactions, and evolving scenes. The increasing prevalence of unified task formulations directly supports the field-wide move toward temporally grounded perception systems.

**Architectural evolution.** Architectural trends reveal a clear dominance of transformer-based models across detection and tracking tasks. Their ability to model long-range dependencies and global context aligns naturally with challenges such as occlusion handling, identity preservation, and multi-view aggregation. Transformer architectures are particularly prevalent in joint detection-tracking, point tracking, and 3D perception.

Diffusion-based models, while still representing a smaller fraction of the literature, exhibit the fastest growth rate, especially in tracking and temporal interpolation tasks. Their emergence signals a broader shift toward generative temporal modeling to address motion uncertainty and sparse or irregular observations. In contrast, state-space and recurrent architectures maintain a strong presence in real-time and resource-constrained applications, underscoring the persistent trade-off between modeling flexibility and computational efficiency.

**Modalities and sensor diversity.** RGB-based perception continues to dominate the literature, but the proportion of multimodal approaches has increased consistently over the surveyed period. LiDAR- and multi-view-based methods are particularly prominent

in 3D detection and tracking, driven by autonomous driving and robotics applications. Thermal, event-based, and audio-visual modalities, while less common, show steady adoption in domain-specific settings where RGB sensing is unreliable.

Vision-language methods constitute the most rapidly expanding subset, enabled by advances in large multimodal models. These approaches support open-vocabulary detection and referring tracking, substantially expanding perceptual flexibility. At the same time, their growing prevalence introduces new dependencies on language quality, semantic alignment, and prompt sensitivity, raising open questions about robustness and evaluation.

**Targeted challenges and research gaps.** An analysis of explicitly addressed challenges shows that occlusion, identity switches, fast or irregular motion, and domain shift are the most frequently targeted problems in recent work. As shown in Figure 2 (also see review table in Appendix for more details), challenges such as small objects, low visibility, and sparse observations also receive substantial attention, particularly in application-driven settings. Broader contextualization of these challenge trends, including how they are framed across surveys on object detection, single- and multi-object tracking, 3D, multi-view, and multimodal perception, vision-language and open-world perception, as well as benchmark-centered analyses, is provided in Appendix. In contrast, relatively few methods directly address uncertainty estimation, annotation efficiency, or long-term open-world scalability, despite their practical importance. This imbalance suggests that certain challenges persist not due to lack of awareness, but due to the difficulty of integrating them into existing architectures and evaluation protocols.

**Temporal dynamics and emerging directions.** Examining trends over time shows several consistent patterns. Transformer-based and joint detection-tracking methods exhibit steady growth, while diffusion-based tracking and interpolation approaches show rapid recent adoption. Multimodal and vision-language methods experience the largest relative increase, reflecting a shift toward more semantically rich and flexible perception systems. Conversely, purely frame-based and task-isolated methods show a gradual decline, indicating a broader paradigm shift toward unified, temporally grounded, and challenge-aware models. These quantitative trends reinforce the qualitative observations made throughout this survey and motivate the open challenges and future directions.

**Datasets and evaluation practices.** Figure 1 highlights that standard benchmarks such as MOT17/20, KITTI, and Waymo, remain central to experimental evaluation, reflecting their role in ensuring comparability and reproducibility. However, the analysis also shows increasing use of 3D, multi-view, RGB-D, thermal, event-based, and synthetic datasets, suggesting a gradual broadening of evaluation settings. Despite this diversification, benchmark concentration remains high, potentially biasing reported progress toward specific scenarios. A similar pattern emerges in evaluation metrics (Figure 3): spatial accuracy metrics (*e.g.*, mAP) dominate detection, while identity-aware metrics (*e.g.*, IDF1, HOTA) dominate tracking. Metrics explicitly capturing long-term robustness, uncertainty awareness, or open-world behavior remain comparatively rare, highlighting a mismatch between evaluation practices and real-world deployment requirements. While standardized metrics such as mAP, MOTA, IDF1, and HOTA have played a crucial role in enabling reproducible benchmarking, each of them captures only

a partial view of system performance. mAP remains highly effective for evaluating spatial localization and category recognition accuracy, yet it neglects temporal coherence and identity preservation, which are fundamental for tracking-oriented applications. Identity-centric measures such as IDF1 emphasize long-term association consistency but may underrepresent localization errors and short-term instabilities. Similarly, MOTA aggregates multiple error sources into a single score, which can obscure the specific failure modes of a system, particularly in crowded or highly dynamic scenes. More recent metrics such as HOTA attempt to provide a balanced assessment of detection and association quality; however, they still do not explicitly account for semantic correctness, uncertainty, or open-world behavior. These complementary strengths and limitations indicate that no single metric is sufficient to characterize real-world performance, underscoring the importance of scenario-aware evaluation protocols that align more closely with deployment requirements.

## 4 Analysis and Discussion

This section synthesizes the surveyed literature through a set of analytical perspectives. Rather than reiterating method-level descriptions, we distill higher-level insights about how architectural choices, supervision paradigms, sensing modalities, and evaluation practices have co-evolved in response to persistent real-world challenges. Grounded in quantitative trends presented earlier, this discussion aims to clarify not only *what* design patterns dominate recent research, but *why* they have emerged and *when* they are most appropriate in practice.

**Challenge-driven convergence of detection and tracking.** One of the most prominent patterns across the literature is the gradual convergence of object detection and tracking, driven less by task unification per se than by shared challenges such as occlusion, identity persistence, fast motion, and long-term temporal consistency. Classical task-centric pipelines, most notably tracking-by-detection, remain widely used due to their modularity and efficiency. However, an increasing number of recent methods reformulate tracking as a temporally extended inference problem, where identities, trajectories, or object states are predicted jointly rather than associated heuristically. This shift is reflected in the rise of joint detection-tracking frameworks, end-to-end identity prediction models, and trajectory-centric formulations. By treating object trajectories as structured temporal entities, these approaches reduce sensitivity to intermittent detection failures and ambiguous associations. Empirical evidence across diverse benchmarks consistently shows that methods explicitly encoding temporal structure, through long-horizon attention, sequence modeling, or generative temporal processes, achieve superior robustness under heavy occlusion and non-linear motion.

> **Key insight.** Performance gains in modern tracking are increasingly determined by how effectively temporal structure is modeled, rather than by incremental improvements in per-frame detection accuracy.

**Architectural trade-offs and practical guidance.** The surveyed literature shows three dominant architectural paradigms

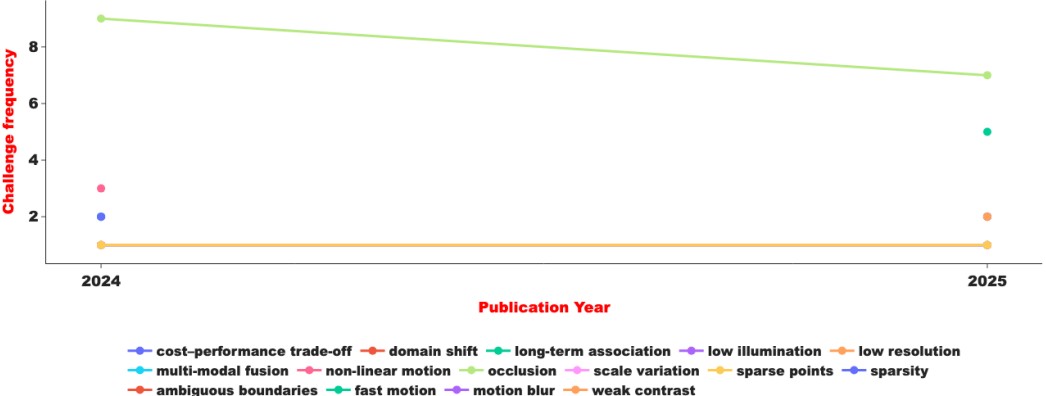

**Figure 2: Temporal trends of prominent research challenges in object detection and tracking (2024-2025). The figure reports the year-wise frequency of the most frequently discussed challenges, manually extracted from the Challenges column of the literature review in Appendix. For clarity, only the top challenges are visualized. This temporal window is intentionally restricted to emphasize the most recent shifts in research focus, while earlier works (2022-2023) are synthesized qualitatively in the taxonomy and related work sections. The observed trends highlight both persistent bottlenecks and emerging difficulties, illustrating how research priorities have evolved in recent literature.**

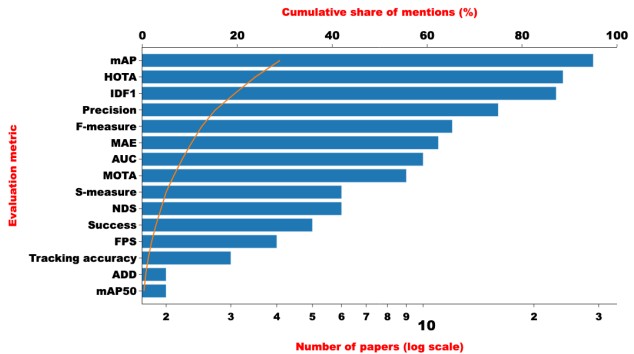

**Figure 3: Evaluation metric usage across recent object detection and tracking literature (2024-2025). Horizontal bars show the raw frequency of metric mentions extracted from the Metrics column of the literature review table in Appendix and plotted on a logarithmic bottom *x*-axis to improve readability across orders of magnitude. The overlaid curve, shown on the top *x*-axis, denotes the cumulative share of mentions. The distribution highlights the predominance of standard multi-object tracking metrics (*e.g.*, HOTA, IDF1, MOTA, *etc.*), alongside the widespread adoption of detection-oriented metrics (*e.g.*, mAP and its variants), reflecting prevailing evaluation practices in the field.**

shaping contemporary detection and tracking systems, each corresponding to different operational regimes and constraints.

Transformer-based architectures have become the default choice for many joint detection-tracking, point tracking, and 3D perception tasks. Their capacity for global context modeling and long-range dependency capture directly addresses challenges such as multi-object interaction, occlusion, and identity switching. At the same time,

their computational cost and data hunger motivate ongoing work on sparse attention, hierarchical representations, and efficiency-oriented variants. Diffusion-based and other generative temporal models represent a rapidly growing, though still emerging, paradigm. By framing motion estimation and temporal interpolation as probabilistic generation processes, these methods offer increased robustness to noise, missing observations, and complex dynamics. Their current limitations, most notably inference latency and training complexity, restrict widespread deployment, but their rapid adoption suggests strong potential for scenarios dominated by uncertainty and sparsity. In contrast, state-space and recurrent models, including modernized filtering-based approaches, remain central in real-time and large-scale applications. Hybrid designs that combine learned visual representations with classical motion models strike a pragmatic balance between interpretability, stability, and efficiency.

> **Practical guidance.** Transformer-based models are most effective when long-term interactions dominate; diffusion-based approaches are well suited to highly uncertain or sparse observations; and state-space hybrids remain preferable under strict real-time or resource constraints.

**Detection paradigms and their impact.** The literature shows that open-vocabulary and open-world detectors, while powerful, frequently degrade temporal stability due to fluctuating category assignments and ambiguous semantic grounding. Successful integration into tracking pipelines therefore requires mechanisms that enforce semantic consistency across time, incorporate uncertainty-aware identity modeling, or jointly reason about category and identity persistence. Object detection research has undergone a parallel transformation, moving beyond closed-set recognition toward open-vocabulary, few-shot, weakly supervised, and domain-generalized formulations. These advances expand category coverage and reduce annotation dependence, often through semantic embeddings,

vision-language alignment, or generative context modeling. However, they also introduce new sources of uncertainty that directly affect downstream tracking.

> **Key insight.** Expanding semantic flexibility in detection without explicit temporal and uncertainty modeling often shifts, rather than resolves, failure modes in tracking systems.

**Multimodal and 3D perception as a unifying trend.** A clear trend across recent work is the expansion from RGB-centric perception toward multimodal and 3D-aware systems. LiDAR, RGB-D, thermal, event-based, audio-visual, and vision-language inputs are increasingly used to address sensing limitations in challenging environments. Despite modality-specific implementations, common issues recur: sensor misalignment, missing data, and heterogeneous noise. Rather than early fusion through feature concatenation, recent methods favor adaptive fusion strategies, including modality gating, uncertainty-aware weighting, and late fusion within shared semantic spaces. In 3D and multi-view settings, temporal aggregation and geometric constraints play a critical role in mitigating sparsity and occlusion, reinforcing the importance of temporal reasoning beyond purely spatial representations.

> **Practical guidance.** Multimodal systems achieve the greatest robustness when fusion strategies are adaptive and uncertainty-aware, allowing the model to selectively rely on the most informative sensors under changing conditions.

**Evaluation practices, biases, and deployment implications.** Our analysis highlights a strong dependence on a small set of benchmarks and metrics, particularly mAP for detection and ID-aware metrics such as HOTA and IDF1 for tracking. While these metrics facilitate comparison, they often fail to capture long-term robustness, semantic correctness, or uncertainty, factors that dominate real-world performance. The proliferation of tasks, modalities, and datasets has further fragmented evaluation practices, making cross-method comparison increasingly difficult. Reported gains on standard benchmarks do not always translate to deployment scenarios involving domain shift, open-set conditions, or sensor degradation.

From a deployment perspective, the literature suggests that no single paradigm dominates universally. Effective system design depends on aligning architectural choices with dominant challenges, data availability, and computational constraints. For example, sports analytics and UAV tracking emphasize fast motion and occlusion, autonomous driving prioritizes 3D reasoning and uncertainty, and vision-language systems trade robustness for semantic flexibility.

> **Key insight.** Benchmark-driven optimization risks obscuring real-world failure modes; progress increasingly depends on challenge-aware evaluation and deployment-oriented design choices.

**Remaining gaps and research opportunities.** Despite substantial progress, several gaps persist across the surveyed literature. Unified detection-tracking systems still struggle with open-world scalability and semantic drift. Diffusion-based models require further optimization for real-time operation. Evaluation protocols lag behind architectural advances, particularly for multimodal and open-world settings. Finally, integrating large multimodal models into efficient, interpretable, and temporally consistent perception pipelines remains an open challenge. Addressing these issues require coordinated advances in architectures, supervision strategies, datasets, and evaluation frameworks, reinforcing the need for challenge-driven perspectives.

## 5 Open Challenges and Future Directions

Despite remarkable progress in object detection and tracking, several fundamental challenges persist. These challenges are not isolated to specific tasks or modalities but stem from broader limitations in how current systems are formulated, trained, and evaluated. In this section, we highlight key open problems and propose promising research directions that can shape the next generation of detection and tracking systems.

**Scalable open-world detection and tracking.** Current detection and tracking systems typically operate in closed or moderately open-world settings, where the set of object categories and operating conditions are predefined. However, real-world applications require systems capable of handling novel objects, evolving appearances, and shifting environments over time. While open-vocabulary detection and referring tracking have made strides toward this goal, existing methods struggle to maintain temporal consistency, identity stability, and uncertainty estimation in open-world settings.

Research should focus on scalable open-world formulations that address category discovery, semantic grounding, and identity management throughout extended temporal spans. Lifelong and continual learning paradigms, together with principled uncertainty estimation and memory management techniques, will be crucial for enabling long-term open-world tracking. Ensuring consistency in identity recognition across novel categories and environments is key for reliable real-world performance.

**Unified evaluation protocols across tasks and modalities.** The rapid proliferation of tasks, datasets, and evaluation metrics has led to fragmented benchmarking practices. Metrics commonly used for detection, single-object tracking, multi-object tracking, and 3D tracking are often incompatible, hindering comprehensive comparisons and making it difficult to track progress in addressing shared challenges like occlusion, identity preservation, and temporal continuity. To address these issues, there is a pressing need for unified, challenge-aware evaluation frameworks that measure not only spatial accuracy but also temporal consistency, semantic correctness, and uncertainty across tasks and modalities. These frameworks should be extensible to open-world and multimodal scenarios, and need to reflect deployment conditions more accurately than traditional idealized benchmarks.

**Advancing long-term temporal reasoning.** Although recent trackers increasingly integrate temporal modeling, the majority of methods focus on short to medium-term associations. Long-term temporal reasoning, such as identity recovery after extended occlusion, reasoning over sparse observations, and understanding object-level temporal semantics, remains a significant challenge.

Approaches should integrate generative temporal models, memory-augmented architectures, and explicit state representations that can persist across long sequences. Bridging low-level motion modeling with higher-level semantic understanding and event-based

reasoning will help achieve a more human-like ability to interpret dynamic scenes over extended periods.

**Efficient integration of large multimodal models.** Large multimodal models, including vision-language foundations, have demonstrated substantial potential in tasks requiring generalization across diverse settings. However, their direct application to detection and tracking remains constrained by computational overhead, latency, and limited interpretability. Moreover, the interaction between language grounding and temporal consistency is still an under-explored area. Research should focus on the efficient integration of large multimodal models into detection and tracking pipelines. This can be achieved through lightweight prompting, modular architectures, and selective semantic reasoning that allows these models to enhance system capabilities without overwhelming computational resources. Such models should be optimized for both open-vocabulary performance and temporal consistency.

**Annotation-efficient.** High-quality annotations are a bottleneck for scaling detection and tracking systems to diverse environments, modalities, and domains. While weakly-supervised and self-supervised approaches have made headway, they still lag behind fully supervised methods, particularly in complex tracking scenarios. Further research is needed to exploit temporal consistency, cross-modal alignment, and physical constraints as supervisory signals. Self-supervised learning from unlabelled video data, combined with minimal human intervention, presents a promising approach to building scalable and adaptable perception systems that are not heavily reliant on large labeled datasets.

**Robustness, uncertainty, and reliability.** As detection and tracking systems are deployed in safety-critical applications, *e.g.*, autonomous vehicles and medical imaging, ensuring robustness and reliability is paramount. Current models tend to produce overconfident predictions and lack mechanisms for expressing uncertainty, particularly when encountering domain shift or sensor degradation. The development of uncertainty-aware detection and tracking frameworks, capable of quantifying confidence at the spatial, temporal, and semantic levels, remains an open challenge. Effective handling of uncertainty is essential for safety, human-machine interaction, and downstream decision-making, especially in real-world, dynamic environments.

**Towards holistic perception systems.** Object detection and tracking should not be viewed in isolation but as integral components of larger perception systems that also include scene understanding, action recognition, and causal reasoning. Future research should aim at integrating these components into systems that are adaptive, interpretable, and context-aware, enabling robust interaction and decision-making across a variety of tasks. Moving toward holistic perception systems requires cross-disciplinary collaboration and a shift toward system-level evaluation. This will involve not only improving individual detection and tracking components but also creating frameworks that integrate reasoning about actions, intents, and causal relationships into the perception pipeline.

## 6 Conclusion

This survey examined recent advances in object detection and tracking (from 2022 to 2025), a period characterized by rapid architectural

innovation, expanding sensing modalities, and a growing emphasis on real-world robustness. By reviewing a broad spectrum of methods spanning 2D and 3D perception, single- and multi-object tracking, multi-view and multimodal sensing, and vision-language integration, we showed that the field is undergoing a fundamental transition: from task-isolated pipelines toward unified, temporally grounded, and semantically enriched perception systems.

A central contribution of this survey is a challenge-centric synthesis of the literature. Rather than organizing prior work around traditional task definitions, we structured the review around the dominant bottlenecks that repeatedly shape performance in practice, including occlusion and long-term identity association, fast and non-linear motion, sparse or degraded observations, domain and category shift, and computational constraints. This perspective shows that many seemingly disparate methods are driven by shared design principles, such as explicit temporal modeling, joint detection-tracking formulations, uncertainty-aware representations, and adaptive multimodal fusion. Framing progress through these challenges provides a more faithful account of how and why recent paradigms have emerged.

Complementing the qualitative analysis, we conducted a quantitative trend analysis that highlights how research priorities are influenced by dataset availability, benchmark selection, and evaluation metrics. While a small number of standard benchmarks and metrics continue to dominate, our analysis shows a clear and accelerating shift toward 3D, multi-view, RGB-D, thermal, event-based, and vision-language datasets, reflecting growing interest in complex and realistic scenarios. At the same time, the lack of unified evaluation protocols across tasks and modalities remains a major barrier to holistic assessment and fair comparison, often obscuring real-world strengths and weaknesses.

The synthesis presented in this survey also exposes several unresolved challenges. Scalable open-world detection and tracking, capable of handling novel categories, evolving environments, and ambiguous supervision over long time horizons, remains largely unsolved. Evaluation practices lag behind architectural advances, particularly with respect to temporal robustness, semantic consistency, and uncertainty. Moreover, while large multimodal and generative models offer powerful new capabilities, their integration into efficient, reliable, and interpretable perception systems suitable for deployment is still an open research problem. Finally, reducing reliance on dense annotations while preserving robustness and generalization is essential for scaling perception systems beyond curated benchmarks. We hope this survey serves not only as a reference, but also as a conceptual framework for understanding recent progress and guiding future research. By shifting the focus from task-specific formulations to the challenges that fundamentally shape performance in real-world environments, this work aims to encourage the development of object detection and tracking systems that are not only more accurate, but also more general, robust, and aligned with the demands of practical deployment.

## Acknowledgments

Arpit Sharma conducted this work while participating remotely in research activities associated with the ARC Research Hub, with his contribution primarily involving a literature review.

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

## A Related Surveys

A large body of survey literature exists on object detection and tracking. However, recent advances between 2022 and 2025 have substantially altered the technical landscape, introducing new paradigms, such as diffusion-based temporal modeling, open-vocabulary perception, vision-language tracking, and unified multi-task architectures, that are not fully captured by earlier reviews. In this section, we position our survey with respect to existing works and clarify how it differs in scope, organization, and analytical depth.

**Surveys on object detection.** Early surveys on object detection primarily focused on convolutional architectures, anchor-based versus anchor-free paradigms, and benchmark-driven performance comparisons on datasets such as PASCAL VOC and MS COCO [75, 90, 121, 199]. More recent reviews have expanded to transformer-based detectors and addressed specialized settings such as small-object detection, aerial imagery, and domain adaptation. However, most existing detection surveys remain task-centric and architecture-driven, treating object detection as a standalone problem and emphasizing incremental model design choices. Open-vocabulary detection [197], domain-generalized detection [140], weakly supervised learning [184, 184], and multimodal detection are often discussed in isolation or omitted entirely. Moreover, detection surveys rarely connect advances in detection to downstream temporal reasoning or tracking [62]. Rather than treating detection as an isolated task, our survey places detection methods, closed-set, open-vocabulary, few-shot, and weakly supervised, within a broader perception pipeline that increasingly overlaps with tracking, semantics, and multimodal reasoning. We explicitly analyze detection research through the challenges it addresses (e.g., domain shift, small objects, low visibility), and we connect detection advances to their implications for long-term tracking and unified detection-tracking models.

**Surveys on multi-object and single-object tracking.** Several surveys have reviewed single-object tracking (SOT) [63] and multi-object tracking (MOT) [86, 129], typically organizing methods into appearance-based, motion-based, or tracking-by-detection frameworks. More recent works discuss transformer-based trackers and benchmark evolution, particularly on datasets such as LaSOT, GOT-10k, and MOT17/20 [93, 146, 158].

Nevertheless, most tracking surveys predate the recent surge of diffusion-based tracking [168], end-to-end identity prediction, state-space models, and generative temporal modeling. Furthermore, SOT [91] and MOT are often reviewed separately, despite increasing architectural convergence [182]. Existing surveys also tend to emphasize algorithmic pipelines while under-analyzing real-world challenges such as long-term occlusion, identity recovery, fast motion, sparse observations, and camera motion [98, 150].

Our survey unifies SOT, MOT, long-term tracking, and point-level tracking under a common analytical framework. By focusing on the challenges that motivate recent designs, rather than pipeline categories, we expose shared principles across diffusion-based trackers, joint detection-tracking models, and unified tracking-with-detection systems. This challenge-driven perspective allows us to identify trends that cut across traditional tracking boundaries.

**Surveys on 3D, multi-view, and multimodal perception.** Recent surveys have explored 3D object detection, multi-view perception, and multimodal sensing (e.g., RGB-D, LiDAR, thermal,

event cameras) [104, 115, 122]. These works typically focus on modality-specific architectures, sensor fusion strategies, or benchmark comparisons in autonomous driving and robotics [57, 78, 137].

However, existing surveys often treat 3D detection, 3D tracking, and multi-view tracking as separate domains, with limited discussion of their shared challenges [58, 73, 190]. Moreover, the growing role of uncertainty modeling, weak supervision, and annotation-efficient learning in 3D perception is rarely integrated into a unified analysis [101, 131]. Vision-language and audio-visual tracking are also largely absent from traditional multimodal surveys [60, 198].

We integrate 2D, 3D, multi-view, and multimodal perception into a single survey, emphasizing how similar challenges, such as occlusion, sparsity, sensor noise, and cross-domain generalization, manifest across modalities. By jointly analyzing multimodal fusion, uncertainty-aware modeling, and annotation-efficient learning, our survey highlights unifying trends that transcend sensor-specific solutions.

**Surveys on vision-language and open-world perception.** With the emergence of large multimodal models, several recent surveys have addressed vision-language learning, open-vocabulary recognition, and foundation models for vision [59, 80, 110, 188]. These surveys typically focus on representation learning, prompting strategies, or high-level reasoning tasks [82, 178].

Yet, relatively few works examine how vision-language models reshape object detection and tracking [179, 196] specifically, particularly in open-world, zero-shot, and referring settings [138]. The interaction between language grounding, temporal association, and identity consistency remains underexplored in survey form [102, 187].

Our survey explicitly incorporates vision-language detection and tracking as first-class components of modern visual perception. We analyze how language supervision alters task definitions, evaluation metrics, and failure modes, and we connect VL-based methods to broader trends such as open-set generalization and semantic trajectory understanding, topics largely absent from existing tracking surveys.

**Meta-analyses and benchmark-centered reviews.** Some surveys focus on benchmark evolution, dataset design, or evaluation metrics, providing valuable historical context [69, 76, 148]. However, such works often lack a holistic view of how datasets, metrics, and modeling choices jointly influence research directions [155]. Quantitative trend analysis across tasks and challenges is rarely performed.

Beyond qualitative discussion, we provide a quantitative meta-analysis of dataset usage, metric prevalence, and challenge frequency from 2022 to 2025. This enables us to empirically identify shifts in research priorities and to reveal biases in current benchmarks, an aspect missing from most existing surveys.

## B Literature Review

This section reviews recent object detection and tracking literature through a structured narrative that complements the comparative analysis presented in Tables 2. Rather than cataloging individual methods, we organize prior work by task formulation and dominant challenges, highlighting recurring design principles, empirical trends, and points of convergence across subfields. This perspective

**Table 2: Recent literature on object detection and tracking, reporting each paper's model category, datasets, metrics, task type, and targeted challenges. Dataset names follow standard abbreviations used in the field, including COCO (Common Objects in Context), MOT17/MOT20 (Multi-Object Tracking benchmarks), LaSOT (Large-Scale Single Object Tracking), GOT-10k (General Object Tracking), TAO (Tracking Any Object), KITTI (Karlsruhe Institute of Technology dataset), VisDrone, ScanNet, and others. Metrics also appear in their common forms: HOTA (Higher-Order Tracking Accuracy), IDF1 (Identity F1 Score), MOTA (Multiple Object Tracking Accuracy), AP/mAP (Average Precision / mean AP), BEV-mAP (Bird's-Eye-View mAP), NDS (nuScenes Detection Score), FPS (Frames Per Second), and F-measure/MAE as applicable. The table highlights key methodological trends such as transformer-based trackers, diffusion models, YOLO-family detectors, 3D perception networks, and vision-language models. Challenges addressed include occlusion, fast motion, small-object detection, low illumination, domain shift, and data imbalance, providing a concise snapshot of current advances in detection and tracking research.**

| Paper | Model & Category | Datasets | Metrics | Task | Challenges |
|---|---|---|---|---|---|
| [79] Multiple Object Tracking as ID Prediction (CVPR 2025) | MOTIP — End-to-end ID-prediction tracker (DETR-based) | DanceTrack, SportsMOT, MOT17 | HOTA, IDF1, MOTA | MOT | ID-association, fast motion |
| [89] Deconfusetrack: Dealing with confusion for multi-object tracking (CVPR 2024) | DeconfuseTrack — Decomposed Data Association + ONMS | MOT17, MOT20 | HOTA, IDF1, AssA | MOT | Confusion, occlusion |
| [164] Multi-object tracking in the dark (CVPR 2024) | LMOT — Low-light MOT dataset + methods | LMOT (new), MOT17 | HOTA, IDF1, MOTA | MOT in low-light | Low illumination, noise |
| [139] Towards generalizable multi-object tracking (CVPR 2024) | GeneralTrack — Tracking-by-detection + relation reasoning | MOT17, MOT20, BDD100K, TAO | HOTA, IDF1, MOTA | MOT | Domain generalization |
| [70] Yolo-world: Real-time open-vocabulary object detection (CVPR 2024) | YOLO-World — Real-time open-vocabulary YOLO detector | COCO, LVIS | mAP, AP_r, FPS | Detection (open-vocab) | Long-tail, novel classes |
| [133] DINTR: Tracking via diffusion-based interpolation (CVPR 2024) | DINTR — Diffusion-based interpolation for tracking | Standard MOT benchmarks | HOTA, IDF1 | MOT | Motion interpolation, sparse frames |
| [109] Beyond mot: Semantic multi-object tracking (ECCV 2024) | SMOT — Semantic Multi-Object Tracking (semantic trajectory labels) | BenSMOT benchmark (semantic MOT) | HOTA, IDF1 | MOT (Semantic MOT) | Semantics + trajectory understanding |
| [105] Taptrv2: Attention-based position update improves tracking any point ( NeurIPS 2024) | TAPTRv2 — Tracking Any Point Transformer (point-level DETR approach) | TAP / DAVIS-style datasets | AUC, Precision | Point tracking | Long-term occlusion robustness |
| [97] Cotracker: It is better to track together (ECCV 2024) | CoTracker — Joint point tracking via transformer | DAVIS / point-tracking benchmarks | Precision, Long-term metrics | Dense point tracking | Occlusion, long-range consistency |
| [159] Omnitracker: Unifying visual object tracking by tracking-with-detection (TPAMI 2024) | OmniTracker — Unified tracking-with-detection model | LaSOT, TrackingNet, DAVIS16–17, MOT17, MOTS20, YTVIS19 | AUC, Precision, J&F, HOTA, IDF1, MOTA, AP | VOT (SOT, VOS, MOT, MOTS, VIS) | Redundant architectures, multi-task fragmentation, parameter overhead |
| [129] Ego-motion aware target prediction module for robust multi-object tracking (CVPR 2024) | EMAP — Ego-motion Aware Target Prediction (KF-based module) | KITTI MOT | HOTA, IDSW | MOT (DBT) | Camera motion disturbance, missing detections, KF velocity-model limitations, identity switches |
| [154] Lifting multi-view detection and tracking to the bird's eye view (CVPR 2024) | Multi-View Aggregation Tracker — BEV lifting + temporal feature aggregation | Wildtrack, MultiviewX, Synthehicle | mAP, HOTA, IDF1 | Multi-view detection + MOT | Occlusion, missed detections, cross-scene generalization, multi-domain tracking |
| [173] Highly efficient and unsupervised framework for moving object detection in satellite videos (TPAMI 2024) | Unsupervised Sparse-Convolution Detector — Sparse spatio-temporal point-cloud SVMOD framework | Satellite video SVMOD datasets | F1, Precision, Recall (SVMOD standard) | SVMOD (Moving object detection in satellite videos) | Small/dim targets, foreground–background imbalance, high annotation cost, computation redundancy |
| [72] Ada-track: End-to-end multi-camera 3d multi-object tracking with alternating detection and association (CVPR 2024) | Multi-camera 3D-MOT architectures | MOT3D, nuScenes variants | 3D-mAP, IDF1 | 3D multi-camera tracking | Cross-camera re-id, occlusion |
| [192] ODOV: Towards Open-Domain Open-Vocabulary Object Detection (CVPR 2025) | ODOV Detector — Domain-aware open-vocabulary object detection | OD-LVIS (18 domains, 1203 categories) | mAP, mAP50, recall | Open-vocabulary object detection | Domain shift, category shift, open-set generalization |
| [65] ReferGPT: Towards Zero-Shot Referring Multi-Object Tracking (CVPR 2025) | ReferGPT — Zero-shot referring multi-object tracker (vision–language) | Refer-KITTI, Refer-KITTIv2, Refer-KITTI+ | HOTA, IDF1 | Referring multi-object tracking (zero-shot VL-MOT) | Zero-shot generalization, open-set text queries, semantic matching |
| [118] ESNet: Evolution and succession network for high-resolution salient object detection (ICML 2024) | Two-Stage HRSOD Network — Low-resolution localization + high-resolution refinement | Five HRSOD datasets | BD-MAE, MAE, F-measure | High-Resolution Salient Object Detection | Detail preservation, high computational cost, resolution imbalance |
| [99] Real-time multi-object detection and tracking in UAV systems: improved YOLOv11-EFAC and optimized tracking algorithms (IJCV 2024) | YOLOv11-EFAC — UAV-oriented real-time small-object detection + tracking framework | COCO + VisDrone + UAVDT hybrid dataset | mAP@0.5, MOTA, IDSW | UAV small-object detection + MOT | Tiny objects, rapid viewpoint changes, strict real-time constraints, non-linear motion |
| [149] Temporal Coherent Object Flow for Multi-Object Tracking (AAAI 2024) | OFTrack — Section-based multi-frame Object Flow Tracker | Standard MOT benchmarks (e.g., MOT17, MOT20, BDD100K) | HOTA, IDF1, MOTA | Multi-object tracking (section-based, multi-frame) | Long-term association, trajectory oscillation, motion ambiguity, multi-frame correlation complexity |
| [181] Contextual object detection with multimodal large language models (IJCV 2025) | ContextDET — Contextual generative multimodal detector (MLLM-based) | CODE benchmark + open-vocab detection datasets | mAP, Recall (open-vocab), Referring segmentation metrics | Contextual object detection (vision–language) | Weak perception, poor context grounding, vocabulary mismatch, multimodal ambiguity |
| [152] Chattracker: Enhancing visual tracking performance via chatting with multimodal large language model (NeurIPS 2024) | ChatTracker — MLLM-based VL tracker | TNL2K, LaSOT, OTB-Lang | AUC, Precision, Success | Vision–language tracking | Ambiguous texts, weak VL alignment, prompt quality |

| Paper | Model & Category | Datasets | Metrics | Task | Challenges |
|---|---|---|---|---|---|
| [68] Generalized Semantic Contrastive Learning via Embedding Side Information for Few-Shot Object Detection (TPAMI 2025) | Generalized FSOD model — Side-information guided feature learning | PASCAL VOC, MS COCO, LVIS V1, FSOD-1K, FSVOD-500 | mAP (novel/base splits) | Few-shot object detection | Feature confusion, limited samples, overfitting |
| [43] Single-model and any-modality for video object tracking (CVPR 2024) | Un-Track — Unified multi-modality tracker (RGB–X) | DepthTrack + 4 multi-modal benchmarks | F-score, Precision/Recall | Multi-modality tracking | Modality heterogeneity, scarce data, missing modalities |
| [126] Vscode: General visual salient and camouflaged object detection with 2d prompt learning (CVPR 2024) | VSCode — Generalist SOD/COD model with 2D prompts | 26 SOD/COD datasets (4 SOD + 3 COD tasks) | F-measure, MAE, S-measure | Salient + camouflaged object detection | Task redundancy, modality differences, domain/task entanglement |
| [128] Diffmot: A real-time diffusion-based multiple object tracker with non-linear prediction (CVPR 2024) | Diffusion-based MOT (real-time) | DanceTrack, SportsMOT | HOTA, FPS | MOT | Non-linear motion, Fast dynamics |
| [194] Nettrack: Tracking highly dynamic objects with a net (CVPR 2024) | NetTrack — dynamic open-world MOT | TAO, BFT, GMOT-40 | HOTA, IDF1 | Open-world MOT | Highly-dynamic objects, Generalization |
| [63] Hiptrack: Visual tracking with historical prompts (CVPR 2024) | Historical-Prompt Siamese tracker | LaSOT, GOT-10k | AUC, Precision | SOT | Appearance change, Occlusion |
| [84] ASCENT: Annotation-free Self-supervised Contrastive Embeddings for 3D Neuron Tracking in Fluorescence Microscopy (ICCV 2025) | ASCENT — Self-supervised 3D neuron tracker | Public + in-house 3D fluorescence microscopy datasets | Tracking accuracy, ID consistency | 3D neuron tracking | No annotations, volumetric complexity |
| [100] Unidet3d: Multi-dataset indoor 3d object detection (AAAI 2025) | UniDet3D — Unified indoor 3D object detector | ScanNet, S3DIS, ARKitScenes, MultiScan, 3RScan, ScanNet++ | mAP25, mAP50 | Indoor 3D object detection | Small datasets, label-space mismatch, limited generalization |
| [151] L4D-Track: Language-to-4D Modeling Towards 6-DoF Tracking and Shape Reconstruction in 3D Point Cloud Stream (CVPR 2024) | Language-to-4D modeling for 6-DoF tracking | 4D / 6-DoF pose datasets | ADD, Tracking accuracy | 6-DoF object tracking | Unknown object pose, Reconstruction |
| [77] Instagen: Enhancing object detection by training on synthetic dataset (CVPR 2024) | Synthetic-data training for detectors | COCO (synthetic augmentations) | mAP | Detection | Domain gap, Synthetic realism |
| [165] Event stream-based visual object tracking: A high-resolution benchmark dataset and a novel baseline (CVPR 2024) | Event-camera high-frame-rate tracking | EventVOT, FE240Hz, VisEvent, COESOT | AUC, Precision | High-speed tracking | Noisy events, low resolution, high-speed constraints |
| [142] MonoDiff: Monocular 3D Object Detection and Pose Estimation with Diffusion Models (CVPR 2024) | MonoDiff — Diffusion-based monocular 3D detector | KITTI, Waymo | mAP3D, BEV-mAP | Monocular 3D object detection | Lack of 3D cues, high uncertainty |
| [120] ESOD: efficient small object detection on high-resolution images (TIP 2024) | ESOD — Efficient small-object detector (feature-level slicing) | VisDrone, UAVDT, TinyPerson | AP | Small-object detection | Sparse targets, high-res cost |
| [169] Mono3DVLT: Monocular-Video-Based 3D Visual Language Tracking (CVPR 2025) | Mono3DVLT-MT — Monocular 3D vision–language tracker | Mono3DVLT-V2X | 3D IoU, Success, Precision | 3D VL tracking (monocular) | No depth sensors, language–3D gap |
| [144] Conflict-alleviated gradient descent for adaptive object detection (IJCAI 2024) | CAGrad — Gradient-harmonized DAOD optimizer | Cross-domain DAOD benchmarks | mAP | DAOD | Gradient conflict, domain shift |
| [143] Teamtrack: A dataset for multi-sport multi-object tracking in full-pitch videos (CVPR 2024) | TeamTrack — Full-pitch sports MOT benchmark | Soccer, basketball, handball videos | HOTA, IDF1 | Sports MOT | Similar appearance, diverse motion |
| [61] Liso: Lidar-only self-supervised 3d object detection (ECCV 2024) | Trajectory-regularized self-training — Self-supervised LiDAR detector training | Unlabeled LiDAR sequences (real-world datasets) | mAP3D, BEV-mAP | LiDAR 3D object detection | No labels, calibration-free pseudo GT |
| [193] Instance tracking in 3D scenes from egocentric videos (CVPR 2024) | IT3DEgo — Egocentric 3D instance tracking benchmark | RGB-D egocentric videos (IT3DEgo dataset) | 3D tracking accuracy, Recall | 3D egocentric instance tracking | Occlusion, pose changes |
| [130] Sam-pm: Enhancing video camouflaged object detection using spatio-temporal attention (CVPR 2024) | SAM-PM — Temporal-consistency module for SAM (VCOD) | VCOD benchmark datasets | F-measure, MAE | Video camouflage object detection | Camouflage, temporal consistency |
| [135] Vasttrack: Vast category visual object tracking (NeurIPS 2024) | VastTrack — Large-scale general tracking benchmark | 50,610 videos, 2,115 categories | AUC, Precision, Success | General visual tracking (vision + VL) | Category diversity, large scale |
| [111] Sampling-resilient multi-object tracking (AAAI 2024) | SOKF — Sampling-resilient MOT tracker (LSTM-KF) | MOT17, DanceTrack, other MOT benchmarks | HOTA, IDF1 | Down-sampled MOT | Sparse observations, non-linear motion |
| [117] FastTrack: A highly efficient and generic GPU-based multi-object tracking method with parallel Kalman filter (IJCV 2024) | PKF / FastTrack — Parallel GPU-based MOT tracker | MOT17, MOT20, KITTI, DanceTrack | HOTA, IDF1, FPS | Large-scale MOT | Non-uniform motion, scalability |
| [44] Diffusiontrack: Point set diffusion model for visual object tracking (CVPR 2024) | DiffusionTrack — Generative diffusion-based tracker | LaSOT, GOT-10k, TrackingNet (SOT benchmarks) | AUC, Precision | SOT | Distractors, appearance variation |
| [64] MLP-DINO: category modeling and query graphing with deep MLP for object detection (IJCAI 2024) | MLP-DINO — Transformer detector with QICS + MLP + GQS | COCO | AP | Object detection | Box-sensitive categories, query imbalance |
| [166] Snida: Unlocking few-shot object detection with non-linear semantic decoupling augmentation (CVPR 2024) | SNIDA — Semantic-guided non-linear instance augmentation for FSOD | PASCAL VOC, MS COCO (few-shot splits) | mAP | Few-shot object detection | Low data, limited diversity |

| Paper | Model & Category | Datasets | Metrics | Task | Challenges |
|---|---|---|---|---|---|
| [119] Revisiting Siamese-Based 3D Single Object Tracking With a Versatile Transformer (TPAMI 2025) | VPTT — Transformer-based 3D point-cloud tracker | KITTI, nuScenes, Waymo | Success, Precision, FPS | 3D SOT | Sparse points, complex motion |
| [95] Exploring enhanced contextual information for video-level object tracking (AAAI 2025) | MCITrack — Video-level tracker with Mamba-based context fusion | LaSOT, GOT-10k, other SOT benchmarks | AUC, AO | SOT | Context loss, long-range dependencies |
| [145] Focusing on Tracks for Online Multi-Object Tracking (CVPR 2025) | TrackTrack — Track-focused online MOT with TPA + TAI | MOT17, MOT20, Dance-Track | HOTA, IDF1 | Online MOT | Occlusion handling, spurious tracks |
| [127] Exploring point-BEV fusion for 3D point cloud object tracking with transformer (TPAMI 2024) | PTTR / PTTR++ — 3D transformer tracker (coarse-to-fine) | KITTI, nuScenes, Waymo | Success, Precision | 3D SOT | Sparsity, motion cues |
| [83] Divert more attention to vision-language object tracking (TPAMI 2024) | Unified VL Tracker — VL-adaptive tracking (ModaMixer) | 23K VL videos | AUC, Precision | VL SOT | Weak VL cues, limited data |
| [147] Prior-free 3D Object Tracking (CVPR 2025) | BIT — Prior-free 3D tracking via iterative geometry + pose refinement | 3D tracking benchmarks | ADD, Pose error | 3D SOT | No priors, geometry refinement |
| [162] Uncertain Object Representation for Image-Based 3D Object Perception (TPAMI 2025) | Uncertain3D — Probabilistic 3D boxes | nuScenes | NDS, AMOTA | 3D detect + MOT | Uncertain localization |
| [112] Patch-level sounding object tracking for audio-visual question answering (AAAI 2025) | PSOT — Patch-level audio-visual tracking for AVQA | AVQA benchmarks | QA accuracy | AVQA | Sound–motion ambiguity |
| [153] What You Have is What You Track: Adaptive and Robust Multimodal Tracking (ICCV 2025) | MM-MoE Tracker — Adaptive multimodal fusion (missing modalities) | 9 multimodal tracking benchmarks | AUC, Precision | Multimodal SOT | Missing data, modality gaps |
| [71] FASTer: Focal token Acquiring-and-Scaling Transformer for Long-term 3D Objection Detection (CVPR 2025) | FASTer — Focal-token 3D temporal detector | Waymo | mAP, NDS | 3D detection | High complexity |
| [185] Safdnet: A simple and effective network for fully sparse 3d object detection (CVPR 2024) | SAFDNet — Fully sparse 3D detector with adaptive diffusion | Waymo, nuScenes, Argoverse2 | mAP | 3D detection | Long-range, sparsity |
| [123] Unbiased faster r-cnn for single-source domain generalized object detection (CVPR 2024) | UFR — Causal domain-generalized detector (SDG) | Five SDG scenes | mAP | Object detection (SDG) | Data bias, confounders |
| [132] Salient object detection in rgb-d videos (TIP 2024) | DCTNet+ — RGB-D video SOD with multi-modal fusion | RDVS (RGB-D VSOD) | F-measure, MAE | RGB-D VSOD | Depth realism, multi-modal fusion |
| [157] Geometry-Aware 3D Salient Object Detection Network (AAAI 2025) | GeoSOD — Superpoint-based 3D saliency | PCSOD | F-measure | 3D SOD | Blurry boundaries |
| [116] Vidsod-100: A new dataset and a baseline model for rgb-d video salient object detection (IJCV 2024) | ATF-Net — RGB-D video saliency fusion | ViDSOD-100, DAVSOD | F-measure | RGB-D VSOD | Multi-modal fusion |
| [94] L4dr: Lidar-4dradar fusion for weather-robust 3d object detection (AAAI 2025) | L4DR — LiDAR–4D radar fusion | VoD, K-Radar | mAP | 3D detection | Weather robustness |
| [136] Harmonizing Attention Fields with Knowledge Distillation for Multi-View 3D Object Detection (CVPR 2025) | HarmonDistill — Query-aligned distillation for MV-3D detection | nuScenes | NDS, mAP | Multi-view 3D detection | Knowledge alignment |
| [74] Salient Object Detection with Dynamic Convolutions (CVPR 2024) | SODDCNet — Dynamic large-kernel CNN for SOD | COCO+OpenImages (pre-train), SOD benchmarks | F-measure, MAE | SOD / VSOD | Scale variation |
| [175] ImOV3D: Learning Open Vocabulary Point Clouds 3D Object Detection from Only 2D Images (NeurIPS 2024) | ImOV3D — 2D-only trained open-vocab 3D detector | SUNRGBD, ScanNet | mAP | OV-3D detection | Modality gap |
| [124] Lion: Linear group rnn for 3d object detection in point clouds (NeurIPS 2024) | LION — Linear RNN–based 3D detector | Waymo, nuScenes, Argoverse2, ONCE | mAP, NDS | 3D detection | Sparse points |
| [88] OpenM3D: Open Vocabulary Multi-view Indoor 3D Object Detection without Human Annotations (ICCV 2025) | OpenM3D — OV multi-view 3D detector (no labels) | ScanNet200, ARKitScenes | mAP | OV 3D detection | Pseudo boxes |
| [186] Voxel mamba: Group-free state space models for point cloud based 3d object detection (NeurIPS 2024) | Voxel Mamba — Group-free SSM voxel encoder | Waymo, nuScenes | mAP, NDS | 3D detection | Spatial proximity |
| [92] Ptt: Point-trajectory transformer for efficient temporal 3d object detection (CVPR 2024) | PTT — Point–trajectory transformer for temporal 3D detection | Waymo | mAP, NDS | Temporal 3D detection | Memory limits |
| [96] Click Crop & Detect: One-Click Offline Annotation for Human-in-the-Loop 3D Object Detection on Point Clouds (CVPR 2024) | CCD — One-click 3D annotation aid | nuScenes, KITTI | mAP | 3D detection/annotation | Label-efficiency |
| [67] FieldMOT: A Field-Registered Multi-Object Tracking for Sports Videos (CVPRW 2025) | Register-then-Track MOT (sports) | Football (synthetic), Street-view | HOTA, IDF1 | Multi-object tracking (broadcast) | Camera switches, Viewpoint changes |
| [150] No Train Yet Gain: Towards Generic Multi-Object Tracking in Sports and Beyond (CVPR 2025) | McByte — mask-propagation MOT | SportsMOT, DanceTrack, SoccerNet-Tracking, MOT17 | HOTA, IDF1, MOTA | Tracking-by-detection (no training) | Fast motion, Occlusion, Camera shifts |

| Paper | Model & Category | Datasets | Metrics | Task | Challenges |
|---|---|---|---|---|---|
| [98] SportMamba: Adaptive Non-Linear Multi-Object Tracking with State Space Models for Team Sports (CVPR 2025) | SportMamba — Adaptive sports MOT | SportsMOT, VIP-HTD | HOTA, IDF1 | MOT | Fast motion, occlusion |
| [27] Diffusiontrack: Diffusion model for multi-object tracking (AAAI 2024) | DiffusionTrack — diffusion-based JDT tracker | MOT17, MOT20, Dance-Track | HOTA, IDF1 | MOT (joint detection–tracking) | Inconsistency, robustness–complexity trade-off |
| [183] TGFormer: Transformer with Track Query Group for Multi-Object Tracking (AAAI 2025) | TGFormer — grouped track-query transformer | MOT17, MOT20, Dance-Track | HOTA, IDF1 | MOT | Occlusion levels, long-term association |
| [176] MM-Tracker: Motion Mamba for UAV-platform Multiple Object Tracking (AAAI 2025) | MM-Tracker — Motion-Mamba UAV MOT model | UAV-MOT datasets (e.g., VisDrone, UAVDT) | HOTA, IDF1, MOTA | UAV MOT | Global+local motion, motion blur handling |
| [167] Smiletrack: Similarity learning for occlusion-aware multiple object tracking (AAAI 2024) | SMILETrack — Siamese similarity-learning MOT tracker | MOT17, MOT20 | HOTA, MOTA | MOT (tracking-by-detection) | Occlusion, similar objects, cost–performance tradeoff |
| [189] Scale Optimization Using Evolutionary Reinforcement Learning for Object Detection on Drone Imagery (AAAI 2024) | SMILETrack — Siamese similarity-learning MOT tracker | MOT17, MOT20 | HOTA, MOTA | MOT (tracking-by-detection) | Occlusion, similar objects, cost–performance tradeoff |
| [161] Depth-aware concealed crop detection in dense agricultural scenes (CVPR 2024) | RISNet — Recurrent iterative RGB-D segmentation model | ACOD-12K (CCD), COD benchmarks | F-measure, MAE, S-measure | Concealed object detection (agri-domain) | Small/dense crops, depth fusion, occlusion |
| [156] Low-light image enhancement framework for improved object detection in fisheye lens datasets (CVPR 2024) | Transformer-based fisheye traffic detector (AICity'24) | AI City Challenge Track 4 (fisheye traffic data) | F1-score | Traffic object detection | Fisheye distortion, blur, low accuracy |
| [87] You Only Look Around: Learning Illumination-Invariant Feature for Low-light Object Detection (NeurIPS 2024) | YOLA — Illumination-invariant detection | Low-light OD datasets | mAP | Detection | Illumination variation |
| [103] Multi-Camera 3D Object Tracking via 3D Point Clouds and Re-Identification (ICCV 2025) | Geometry-aware 3D MTMC tracker (Transformer + ESC) | AI City Challenge 2025 (3D MTMC) | 3D HOTA | 3D multi-camera tracking | Occlusion, sparse views, rare-object imbalance |
| [81] SO-YOLOv8: A novel deep learning-based approach for small object detection with YOLO beyond COCO (ECA 2025) | SO-YOLOv8 — Small-object enhanced YOLOv8 | PASCAL VOC 2012 | Precision, mAP | Small-object detection | Low resolution, occlusion, scale variation |
| [141] Tracknetv4: Enhancing fast sports object tracking with motion attention maps (ICASSP 2025) | TrackNetV4 — Motion-aware sports ball tracker | Tennis ball, Shuttlecock datasets | Precision, Tracking accuracy | High-speed small-object tracking | Motion blur, occlusion, weak visual cues |
| [113] Sood++: Leveraging unlabeled data to boost oriented object detection (TPAMI 2025) | SOOD++ — Semi-supervised oriented detector | DOTA-V1.5, DOTA-V2.0 | mAP | Oriented object detection (aerial) | Small scale, arbitrary orientation, dense layout |
| [195] Spatial residual for underwater object detection (TPAMI 2025) | BSR5 — Spatial Residual backbone for underwater detection | RUOD (underwater detection) | mAP / AP | Underwater object detection | Feature drift, degradation, low visibility |
| [180] Wholly-WOOD: Wholly Leveraging Diversified-quality Labels for Weakly-supervised Oriented Object Detection (TPAMI 2025) | Wholly-WOOD — Weakly supervised oriented detector | Remote sensing OOD datasets | mAP | Oriented object detection | Weak labels (points/HBoxes), no RBox annotation |
| [85] Weakly supervised monocular 3D object detection by spatial-temporal view consistency (TPAMI 2024) | Weakly-supervised monocular 3D detector (2D-only training) | KITTI / nuScenes (monocular 3D) | mAP, BEV-AP | Monocular 3D detection | No 3D labels, spatial–temporal view consistency |
| [172] Ktcn: Enhancing open-world object detection with knowledge transfer and class-awareness neutralization (IJCAI 2024) | KTCN — SAM-based OWOD with DMLA | COCO, VOC (OWOD) | U-Recall, mAP | Open-world detection | Unknown pseudo-labeling, class bias |
| [106] Lmm-det: Make large multimodal models excel in object detection (ICCV 2025) | LMM-Det — Detector-free multimodal LMM detection | COCO, OD benchmarks | mAP, Recall | Object detection (LMM-based) | Low recall, instruction tuning, inference optimization |
| [134] Cholectrack20: A multi-perspective tracking dataset for surgical tools (CVPR 2025) | CholecTrack20 — Surgical tool tracking benchmark | CholecTrack20 (20 surgical videos) | HOTA, IDF1 | Surgical multi-tool tracking | Occlusion, smoke/bleeding, out-of-view tools |
| [177] Escnet: Edge-semantic collaborative network for camouflaged object detection (ICCV 2025) | ESCNet — Edge–texture coupled COD model | COD benchmarks (CAMO, CHAMELEON, COD10K) | F-measure, MAE, S-measure | Camouflaged object detection | Ambiguous boundaries, weak edges/texture cues |
| [66] Just a hint: Point-supervised camouflaged object detection (ECCV 2024) | Point-COD — One-point supervised COD model | COD10K, CAMO, CHAMELEON | F, MAE, S-measure | Weak COD | Minimal labels, weak boundaries |
| [191] Referring camouflaged object detection (TPAMI 2025) | R2CNet — Referring camouflaged object detector | R2C7K (Ref-COD dataset) | F-measure, S-measure, MAE | Referring COD | Specified-object cues, weak contrast, ambiguous boundaries |
| [170] Uncertainty-aware transformer for referring camouflaged object detection (TIP 2025) | UAT — Uncertainty-aware transformer for Ref-COD | Ref-COD benchmark | F-measure, S-measure, MAE | Referring COD | Feature mismatch, uncertainty, weak contrast |

| Paper | Model & Category | Datasets | Metrics | Task | Challenges |
|---|---|---|---|---|---|
| [114] Gradient-Reweighted Adversarial Camouflage for Physical Object Detection Evasion (ICCV 2025) | GRAC — Robust multi-view adversarial camouflage | Sim/real OD attack benchmarks | ASR, mAP drop | Adversarial attacks | Viewpoint, lighting, texture issues |
| [163] Object detection using event camera: A moe heat conduction based detector and a new benchmark dataset (CVPR 2025) | MoE-HCO — MoE heat-conduction event detector | EvDET200K | mAP, IoU-based metrics | Event-based object detection | Low-light, motion blur, fast-motion |
| [108] Pseudo Visible Feature Fine-Grained Fusion for Thermal Object Detection (CVPR 2025) | PFGF — Fine-grained thermal–visible fusion for detection | Thermal OD benchmarks (T2V-based) | mAP, mAR | Thermal object detection | Weak cross-modality fusion, granularity mismatch, limited visible cues |
| [107] Fd2-net: Frequency-driven feature decomposition network for infrared-visible object detection (AAAI 2025) | FD2-Net — Frequency-driven IR–visible fusion detector | LLVIP, FLIR, M3FD | mAP | Infrared–visible object detection (IVOD) | High/low-frequency mismatch, lost details, weak multimodal coupling |
| [160] Alignment-Free RGB-T Salient Object Detection: A Large-scale Dataset and Progressive Correlation Network (AAAI 2025) | PCNet + UVT20K — Alignment-free RGB-T SOD | UVT20K, multiple RGB-T SOD benchmarks | mAP, F-measure, MAE | RGB-T Salient Object Detection | Unaligned modalities, low illumination, cluttered scenes, complex object structures |

**Table 3: Glossary of commonly used evaluation metrics in object detection and multi-object tracking.**

| Abbreviation | Full Name | What It Measures | Primary Focus | Typical Use |
|---|---|---|---|---|
| mAP | mean Average Precision | Localization and classification accuracy across IoU thresholds | Spatial accuracy | Object detection |
| AP | Average Precision | Precision-recall trade-off for a single class or category | Spatial accuracy | Object detection |
| AR | Average Recall | Coverage of ground-truth objects by predictions | Sensitivity | Object detection |
| MOTA | Multiple Object Tracking Accuracy | Aggregate tracking errors (false positives, false negatives, identity switches) | Overall tracking quality | Multi-object tracking |
| MOTP | Multiple Object Tracking Precision | Localization precision of predicted tracks | Spatial precision | Multi-object tracking |
| IDF1 | Identity F1 Score | Consistency of object identity assignments over time | Temporal consistency | Multi-object tracking |
| IDP | Identity Precision | Correctness of predicted identity associations | Identity correctness | Multi-object tracking |
| IDR | Identity Recall | Recovery of ground-truth identities | Identity robustness | Multi-object tracking |
| HOTA | Higher Order Tracking Accuracy | Joint detection and association performance | Balanced tracking quality | Multi-object tracking |
| DetA | Detection Accuracy | Detection correctness in tracking scenarios | Spatial accuracy | Multi-object tracking |
| AssA | Association Accuracy | Temporal continuity of tracks | Identity association | Multi-object tracking |
| sMOTSA | Soft Multiple Object Tracking and Segmentation Accuracy | Soft matching of detections and tracks | Joint detection-tracking | Multi-object tracking |
| AMOTA | Average MOTA | MOTA averaged over multiple thresholds | Stability | Multi-object tracking |
| CLEAR MOT | Classification of Events, Activities and Relationships (MOT metrics) | Standard MOT error aggregation framework | Aggregate errors | Multi-object tracking |

shows how detection and tracking research is increasingly driven by real-world constraints, such as long-term temporal consistency, openness, and multimodal uncertainty, rather than narrowly defined benchmark tasks.

**Multi-object tracking: evolving beyond association.** MOT remains a central focus of recent research, with a clear shift away from classical tracking-by-detection pipelines toward models that explicitly reason over time. Earlier work primarily targeted improved appearance representations and data association heuristics, but contemporary approaches increasingly emphasize sequence-level modeling to address occlusion, identity switches, and irregular motion. Transformer-based trackers and track-query formulations exemplify this trend by enabling global reasoning across objects and frames, reducing reliance on greedy or local association strategies.

In parallel, generative and interpolation-based approaches, such as diffusion-driven trajectory modeling and multi-frame prediction, have emerged to handle sparse observations and ambiguous motion patterns. Despite these advances, hybrid designs that combine learned representations with classical filtering or motion models remain competitive, particularly in efficiency-critical scenarios such

as UAV tracking and large-scale video analytics. Across benchmarks including MOT17/20, DanceTrack, and SportsMOT, identity preservation under heavy interaction consistently emerges as the dominant performance bottleneck, reinforcing the importance of long-term temporal reasoning over frame-local optimization.

**Single-object and long-term tracking: from similarity to memory.** SOT has undergone a parallel evolution, driven by the need to handle long-term appearance variation, background distractors, and prolonged occlusion. While Siamese architectures remain influential due to their efficiency and simplicity, recent methods increasingly incorporate transformers, memory modules, and historical prompts to capture richer temporal context. This shift reflects an implicit recognition that instantaneous appearance similarity is insufficient for sustained tracking in unconstrained environments.

Long-term tracking benchmarks such as LaSOT, GOT-10k, and TrackingNet consistently show that performance gains are tied to a model's ability to accumulate and selectively reuse historical information. Point-level and dense tracking methods further generalize

this setting by tracking arbitrary visual elements rather than predefined objects, emphasizing temporal consistency over categorical semantics. As a result, the boundaries between SOT, video object segmentation, and even multi-object tracking are becoming increasingly porous, with shared architectures and evaluation principles emerging across tasks.

**Detection beyond the closed set and its impact on tracking.** Object detection research has expanded well beyond the traditional closed-set paradigm, motivated by the need for greater flexibility and reduced annotation dependence. Open-vocabulary detection, few-shot learning, weak supervision, and domain generalization have become prominent themes, often enabled by semantic embeddings, vision-language pretraining, or synthetic data augmentation. Transformer-based detectors and multimodal large models provide powerful mechanisms for category generalization, but they also introduce semantic ambiguity and confidence instability.

These challenges have direct implications for tracking. Open-vocabulary and domain-aware detectors broaden the range of detectable objects but complicate identity assignment and temporal consistency, particularly in open-world settings where category boundaries are fluid. Similarly, few-shot and weakly supervised detectors trade annotation efficiency for increased uncertainty, which can propagate through downstream tracking pipelines. The literature increasingly reflects this interdependence, with detection and tracking being co-designed rather than treated as isolated components.

**3D, multi-view, and temporally grounded perception.** A substantial body of work focuses on 3D object detection and tracking, particularly in autonomous driving, robotics, and indoor scene understanding. These methods span monocular, LiDAR-based, and multi-view configurations, with an increasing emphasis on jointly modeling spatial geometry and temporal dynamics. Transformers and state-space models are commonly employed to aggregate information across viewpoints and time, enabling more robust reasoning under occlusion and sparse sensing.

Key challenges in this domain include incomplete point clouds, sensor noise, and cross-camera identity consistency. As reflected in the surveyed literature, purely spatial reasoning is insufficient for reliable 3D perception; instead, temporal aggregation and motion-aware representations play a decisive role. Uncertainty-aware modeling has also gained prominence, especially in safety-critical applications, where reliable confidence estimation is as important as raw detection accuracy.

**Multimodal and vision-Language paradigms.** Beyond RGB-centric perception, recent work increasingly explores multimodal detection and tracking using depth, thermal imaging, event cameras, audio, and language. These modalities offer complementary cues that improve robustness under challenging conditions such as low illumination, fast motion, or visual camouflage. However, they also introduce practical challenges related to modality alignment, missing data, and heterogeneous noise characteristics.

Vision-language models represent a particularly influential direction, enabling referring expression tracking, zero-shot detection, and open-set perception. While these systems provide strong semantic grounding, they remain sensitive to prompt formulation and linguistic ambiguity, and their temporal reasoning capabilities are often limited. The literature reflects growing interest in tighter integration between visual, temporal, and linguistic representations, suggesting that multimodality is most effective when coupled with explicit temporal modeling.

**Application-driven methods and dataset Trends.** A significant portion of recent research is motivated by domain-specific applications, including UAV imagery, sports analytics, satellite video analysis, medical imaging, underwater exploration, and camouflaged object detection. These domains amplify classical challenges such as small object size, rapid motion, low contrast, and limited annotations, often under strict computational or real-time constraints. Rather than introducing entirely new paradigms, most application-driven methods adapt core detection and tracking principles through domain-aware priors, architectural simplifications, or specialized training strategies.

As summarized in Figures 1 and 3, standard benchmarks such as COCO, MOT17/20, and KITTI remain widely used, but there is a clear diversification toward 3D, multi-view, RGB-D, thermal, event-based, and vision-language datasets. Evaluation metrics increasingly emphasize identity consistency and temporal coherence, reflecting the field's growing focus on long-term robustness. At the same time, the proliferation of task-specific metrics complicates cross-method comparison, underscoring the need for more unified and challenge-oriented evaluation protocols.

## C   Evaluation Metrics Glossary

To improve clarity and accessibility, we summarize the most commonly used evaluation metrics in object detection and tracking in Table 3, including their full names, primary focus, and typical use cases. This glossary serves as a reference for readers, given the diversity and partial overlap of evaluation practices across the literature.

## References for Appendix

[57] Simegnew Yihunie Alaba, Ali C Gurbuz, and John E Ball. 2024. Emerging trends in autonomous vehicle perception: Multimodal fusion for 3D object detection. *World Electric Vehicle Journal* 15, 1 (2024), 20.

[58] Eduardo Arnold, Omar Y Al-Jarrah, Mehrdad Dianati, Saber Fallah, David Oxtoby, and Alex Mouzakitis. 2019. A survey on 3d object detection methods for autonomous driving applications. *IEEE Transactions on Intelligent Transportation Systems* 20, 10 (2019), 3782–3795.

[59] Muhammad Awais, Muzammal Naseer, Salman Khan, Rao Muhammad Anwer, Hisham Cholakkal, Mubarak Shah, Ming-Hsuan Yang, and Fahad Shahbaz Khan. 2025. Foundation models defining a new era in vision: a survey and outlook. *IEEE Transactions on Pattern Analysis and Machine Intelligence* (2025).

[60] Tadas Baltrušaitis, Chaitanya Ahuja, and Louis-Philippe Morency. 2018. Multimodal machine learning: A survey and taxonomy. *IEEE transactions on pattern analysis and machine intelligence* 41, 2 (2018), 423–443.

[61] Stefan Andreas Baur, Frank Moosmann, and Andreas Geiger. 2024. Liso: Lidar-only self-supervised 3d object detection. In *European Conference on Computer Vision*. Springer, 253–270.

[62] Sara Bouraya Jr and Abdessamad Belangour. 2021. Multi object tracking: a survey. In *Thirteenth International Conference on Digital Image Processing (ICDIP 2021)*, Vol. 11878. SPIE, 142–152.

[63] Wenrui Cai, Qingjie Liu, and Yunhong Wang. 2024. Hiptrack: Visual tracking with historical prompts. In *Proceedings of the IEEE/CVF Conference on Computer Vision and Pattern Recognition*. 19258–19267.

[64] Guiping Cao, Wenjian Huang, Xiangyuan Lan, Jianguo Zhang, Dongmei Jiang, and Yaowei Wang. 2024. MLP-DINO: category modeling and query graphing with deep MLP for object detection. In *Proceedings of the Thirty-Third International Joint Conference on Artificial Intelligence (IJCAI-24), Jeju, Republic of Korea*. 3–9.

[65] Tzoulio Chamiti, Leandro Di Bella, Adrian Munteanu, and Nikos Deligiannis. 2025. ReferGPT: Towards Zero-Shot Referring Multi-Object Tracking. In *Proceedings of the Computer Vision and Pattern Recognition Conference*. 3849–3858.

[66] Huafeng Chen, Dian Shao, Guangqian Guo, and Shan Gao. 2024. Just a hint: Point-supervised camouflaged object detection. In *European Conference on Computer Vision*. Springer, 332–348.

[67] Hong-Qi Chen, Chao-Chi Liao, Yuan-Heng Sun, Cheng-Kuan Lin, and Yu-Chee Tseng. 2025. FieldMOT: A Field-Registered Multi-Object Tracking for Sports Videos. In *Proceedings of the Computer Vision and Pattern Recognition Conference*. 5894–5904.

[68] Ruoyu Chen, Hua Zhang, Jingzhi Li, Li Liu, Zhen Huang, and Xiaochun Cao. 2025. Generalized Semantic Contrastive Learning via Embedding Side Information for Few-Shot Object Detection. *IEEE Transactions on Pattern Analysis and Machine Intelligence* (2025).

[69] Wei Chen, Jinjin Luo, Fan Zhang, and Zijian Tian. 2024. A review of object detection: Datasets, performance evaluation, architecture, applications and current trends. *Multimedia Tools and Applications* 83, 24 (2024), 65603–65661.

[70] Tianheng Cheng, Lin Song, Yixiao Ge, Wenyu Liu, Xinggang Wang, and Ying Shan. 2024. Yolo-world: Real-time open-vocabulary object detection. In *Proceedings of the IEEE/CVF conference on computer vision and pattern recognition*. 16901–16911.

[71] Chenxu Dang, ZaiPeng Duan, Pei An, Xinmin Zhang, Xuzhong Hu, and Jie Ma. 2025. FASTer: Focal token Acquiring-and-Scaling Transformer for Long-term 3D Objection Detection. In *Proceedings of the Computer Vision and Pattern Recognition Conference*. 17029–17038.

[72] Shuxiao Ding, Lukas Schneider, Marius Cordts, and Juergen Gall. 2024. Ada-track: End-to-end multi-camera 3d multi-object tracking with alternating detection and association. In *Proceedings of the IEEE/CVF Conference on Computer Vision and Pattern Recognition*. 15184–15194.

[73] Chenjie Du, Chenwei Lin, Ran Jin, Bencheng Chai, Yingbiao Yao, and Siyu Su. 2024. Exploring the state-of-the-art in multi-object tracking: A comprehensive survey, evaluation, challenges, and future directions. *Multimedia tools and applications* 83, 29 (2024), 73151–73189.

[74] Rohit Venkata Sai Dulam and Chandra Kambhamettu. 2025. Salient Object Detection with Dynamic Convolutions. In *Proceedings of the Computer Vision and Pattern Recognition Conference*. 1692–1702.

[75] Okan DURUSOY et al. 2025. Open-Source Datasets for Image Processing and Artificial Intelligence Research: A Comparison of ImageNet and MS COCO Datasets. *International Journal of Sciences and Innovation Engineering* 2, 5 (2025), 639–653.

[76] Mark Everingham, SM Ali Eslami, Luc Van Gool, Christopher KI Williams, John Winn, and Andrew Zisserman. 2015. The pascal visual object classes challenge: A retrospective. *International journal of computer vision* 111, 1 (2015), 98–136.

[77] Chengjian Feng, Yujie Zhong, Zequn Jie, Weidi Xie, and Lin Ma. 2024. Instagen: Enhancing object detection by training on synthetic dataset. In *Proceedings of the IEEE/CVF Conference on Computer Vision and Pattern Recognition*. 14121–14130.

[78] Di Feng, Christian Haase-Schütz, Lars Rosenbaum, Heinz Hertlein, Claudius Glaeser, Fabian Timm, Werner Wiesbeck, and Klaus Dietmayer. 2020. Deep multi-modal object detection and semantic segmentation for autonomous driving: Datasets, methods, and challenges. *IEEE Transactions on Intelligent Transportation Systems* 22, 3 (2020), 1341–1360.

[79] Ruopeng Gao, Ji Qi, and Limin Wang. 2025. Multiple Object Tracking as ID Prediction. In *Proceedings of the IEEE/CVF Conference on Computer Vision and Pattern Recognition*. 27883–27893.

[80] Akash Ghosh, Arkadeep Acharya, Sriparna Saha, Vinija Jain, and Aman Chadha. 2024. Exploring the frontier of vision-language models: A survey of current methodologies and future directions. *arXiv preprint arXiv:2404.07214* (2024).

[81] Kaisar Javeed Giri et al. 2025. SO-YOLOv8: A novel deep learning-based approach for small object detection with YOLO beyond COCO. *Expert Systems with Applications* 280 (2025), 127447.

[82] Jindong Gu, Zhen Han, Shuo Chen, Ahmad Beirami, Bailan He, Gengyuan Zhang, Ruotong Liao, Yao Qin, Volker Tresp, and Philip Torr. 2023. A systematic survey of prompt engineering on vision-language foundation models. *arXiv preprint arXiv:2307.12980* (2023).

[83] Mingzhe Guo, Zhipeng Zhang, Liping Jing, Haibin Ling, and Heng Fan. 2024. Divert more attention to vision-language object tracking. *IEEE Transactions on Pattern Analysis and Machine Intelligence* 46, 12 (2024), 8600–8618.

[84] Haejun Han and Hang Lu. 2025. ASCENT: Annotation-free Self-supervised Contrastive Embeddings for 3D Neuron Tracking in Fluorescence Microscopy. In *Proceedings of the IEEE/CVF International Conference on Computer Vision*. 14676–14687.

[85] Wencheng Han, Runzhou Tao, Haibin Ling, and Jianbing Shen. 2024. Weakly supervised monocular 3D object detection by spatial-temporal view consistency. *IEEE Transactions on Pattern Analysis and Machine Intelligence* (2024).

[86] Saif Hassan, Ghulam Mujtaba, Asif Rajput, and Noureen Fatima. 2024. Multi-object tracking: a systematic literature review. *Multimedia Tools and Applications* 83, 14 (2024), 43439–43492.

[87] Mingbo Hong, Shen Cheng, Haibin Huang, Haoqiang Fan, and Shuaicheng Liu. 2024. You Only Look Around: Learning Illumination-Invariant Feature for Low-light Object Detection. *Advances in Neural Information Processing Systems* 37 (2024), 87136–87158.

[88] Peng-Hao Hsu, Ke Zhang, Fu-En Wang, Tao Tu, Ming-Feng Li, Yu-Lun Liu, Albert YC Chen, Min Sun, and Cheng-Hao Kuo. 2025. OpenM3D: Open Vocabulary Multi-view Indoor 3D Object Detection without Human Annotations. In *Proceedings of the IEEE/CVF International Conference on Computer Vision*. 8688–8698.

[89] Cheng Huang, Shoudong Han, Mengyu He, Wenbo Zheng, and Yuhao Wei. 2024. Deconfusetrack: Dealing with confusion for multi-object tracking. In *Proceedings of the IEEE/CVF Conference on Computer Vision and Pattern Recognition*. 19290–19299.

[90] Jonathan Huang, Vivek Rathod, Chen Sun, Menglong Zhu, Anoop Korattikara, Alireza Fathi, Ian Fischer, Zbigniew Wojna, Yang Song, Sergio Guadarrama, et al. 2017. Speed/accuracy trade-offs for modern convolutional object detectors. In *Proceedings of the IEEE conference on computer vision and pattern recognition*. 7310–7311.

[91] Ke Huang, Zhaoguo Zhang, Jinlong Chen, and Yangying Kou. 2025. A Survey of Single Object Tracking. In *2025 6th International Conference on Electronic Communication and Artificial Intelligence (ICECAI)*. IEEE, 291–299.

[92] Kuan-Chih Huang, Weijie Lyu, Ming-Hsuan Yang, and Yi-Hsuan Tsai. 2024. Ptt: Point-trajectory transformer for efficient temporal 3d object detection. In *Proceedings of the IEEE/CVF Conference on Computer Vision and Pattern Recognition*. 14938–14947.

[93] Xiaonan Huang, Ning Bi, and Jun Tan. 2022. Visual transformer-based models: A survey. In *International Conference on Pattern Recognition and Artificial Intelligence*. Springer, 295–305.

[94] Xun Huang, Ziyu Xu, Hai Wu, Jinlong Wang, Qiming Xia, Yan Xia, Jonathan Li, Kyle Gao, Chenglu Wen, and Cheng Wang. 2025. L4dr: Lidar-4dradar fusion for weather-robust 3d object detection. In *Proceedings of the AAAI Conference on Artificial Intelligence*, Vol. 39. 3806–3814.

[95] Ben Kang, Xin Chen, Simiao Lai, Yang Liu, Yi Liu, and Dong Wang. 2025. Exploring enhanced contextual information for video-level object tracking. In *Proceedings of the AAAI Conference on Artificial Intelligence*, Vol. 39. 4194–4202.

[96] Nitin Kumar Saravana Kannan, Matthias Reuse, and Martin Simon. 2025. Click Crop & Detect: One-Click Offline Annotation for Human-in-the-Loop 3D Object Detection on Point Clouds. In *Proceedings of the IEEE/CVF Conference on Computer Vision and Pattern Recognition*. 4514–4525.

[97] Nikita Karaev, Ignacio Rocco, Benjamin Graham, Natalia Neverova, Andrea Vedaldi, and Christian Rupprecht. 2024. Cotracker: It is better to track together. In *European conference on computer vision*. Springer, 18–35.

[98] Dheeraj Khanna, Jerrin Bright, Yuhao Chen, and John Zelek. 2025. SportMamba: Adaptive Non-Linear Multi-Object Tracking with State Space Models for Team Sports. In *Proceedings of the Computer Vision and Pattern Recognition Conference*. 6143–6153.

[99] Rabia Kıratlı and Alperen Eroğlu. 2025. Real-time multi-object detection and tracking in UAV systems: improved YOLOv11-EFAC and optimized tracking algorithms. *Journal of Real-Time Image Processing* 22, 5 (2025), 1–28.

[100] Maksim Kolodiazhnyi, Anna Vorontsova, Matvey Skripkin, Danila Rukhovich, and Anton Konushin. 2025. Unidet3d: Multi-dataset indoor 3d object detection. In *Proceedings of the AAAI Conference on Artificial Intelligence*, Vol. 39. 4365–4373.

[101] Palvadi Srinivas Kumar. 2025. Uncertainty Estimation in Deep Learning Based Computer Vision. *Applied Computer Vision through Artificial Intelligence* (2025), 331–348.

[102] Xiaohan Lan, Yitian Yuan, Xin Wang, Zhi Wang, and Wenwu Zhu. 2023. A survey on temporal sentence grounding in videos. *ACM Transactions on Multimedia Computing, Communications and Applications* 19, 2 (2023), 1–33.

[103] Jaewon Lee, Heecheol Kim, Doohee Lee, and Kanghee Lee. 2025. Multi-Camera 3D Object Tracking via 3D Point Clouds and Re-Identification. In *Proceedings of the IEEE/CVF International Conference on Computer Vision*. 5358–5365.

[104] Yinjie Lei, Zixuan Wang, Feng Chen, Guoqing Wang, Peng Wang, and Yang Yang. 2023. Recent advances in multi-modal 3d scene understanding: A comprehensive survey and evaluation. *arXiv preprint arXiv:2310.15676* (2023).

[105] Hongyang Li, Hao Zhang, Shilong Liu, Zhaoyang Zeng, Feng Li, Bohan Li, Tianhe Ren, and Lei Zhang. 2024. Taptrv2: Attention-based position update improves tracking any point. *Advances in Neural Information Processing Systems* 37 (2024), 101074–101095.

[106] Jincheng Li, Chunyu Xie, Ji Ao, Dawei Leng, and Yuhui Yin. 2025. Lmm-det: Make large multimodal models excel in object detection. In *Proceedings of the IEEE/CVF International Conference on Computer Vision*. 308–318.

[107] Ke Li, Di Wang, Zhangyuan Hu, Shaofeng Li, Weiping Ni, Lin Zhao, and Quan Wang. 2025. Fd2-net: Frequency-driven feature decomposition network for infrared-visible object detection. In *Proceedings of the AAAI Conference on Artificial Intelligence*, Vol. 39. 4797–4805.

[108] Ting Li, Mao Ye, Tianwen Wu, Nianxin Li, Shuaifeng Li, Song Tang, and Luping Ji. 2025. Pseudo Visible Feature Fine-Grained Fusion for Thermal Object Detection. In *Proceedings of the Computer Vision and Pattern Recognition Conference*. 6710–6719.

[109] Yunhao Li, Qin Li, Hao Wang, Xue Ma, Jiali Yao, Shaohua Dong, Heng Fan, and Libo Zhang. 2024. Beyond mot: Semantic multi-object tracking. In *European*

Conference on Computer Vision. Springer, 276–293.

[110] Zongxia Li, Xiyang Wu, Hongyang Du, Huy Nghiem, and Guangyao Shi. 2025. Benchmark evaluations, applications, and challenges of large vision language models: A survey. *arXiv preprint arXiv:2501.02189* 1 (2025).

[111] Zepeng Li, Dongxiang Zhang, Sai Wu, Mingli Song, and Gang Chen. 2024. Sampling-resilient multi-object tracking. In *Proceedings of the AAAI Conference on Artificial Intelligence*, Vol. 38. 3297–3305.

[112] Zhangbin Li, Jinxing Zhou, Jing Zhang, Shengeng Tang, Kun Li, and Dan Guo. 2025. Patch-level sounding object tracking for audio-visual question answering. In *Proceedings of the AAAI Conference on Artificial Intelligence*, Vol. 39. 5075–5083.

[113] Dingkang Liang, Wei Hua, Chunsheng Shi, Zhikang Zou, Xiaoqing Ye, and Xiang Bai. 2025. Sood++: Leveraging unlabeled data to boost oriented object detection. *IEEE Transactions on Pattern Analysis and Machine Intelligence* (2025).

[114] Jiawei Liang, Siyuan Liang, Tianrui Lou, Ming Zhang, Wenjin Li, Dunqiu Fan, and Xiaochun Cao. 2025. Gradient-Reweighted Adversarial Camouflage for Physical Object Detection Evasion. In *Proceedings of the IEEE/CVF International Conference on Computer Vision*. 13880–13889.

[115] Wei Liang, Pengfei Xu, Ling Guo, Heng Bai, Yang Zhou, and Feng Chen. 2021. A survey of 3D object detection. *Multimedia Tools and Applications* 80, 19 (2021), 29617–29641.

[116] Junhao Lin, Lei Zhu, Jiaxing Shen, Huazhu Fu, Qing Zhang, and Liansheng Wang. 2024. Vidsod-100: A new dataset and a baseline model for rgb-d video salient object detection. *International Journal of Computer Vision* 132, 11 (2024), 5173–5191.

[117] Chongwei Liu, Haojie Li, and Zhihui Wang. 2024. FastTrack: A highly efficient and generic GPU-based multi-object tracking method with parallel Kalman filter. *International Journal of Computer Vision* 132, 5 (2024), 1463–1483.

[118] Hongyu Liu, Runmin Cong, Hua Li, Qianqian Xu, Qingming Huang, and Wei Zhang. 2024. ESNet: Evolution and succession network for high-resolution salient object detection. In *Forty-first International Conference on Machine Learning*.

[119] Jiaming Liu, Yue Wu, Qiguang Miao, Maoguo Gong, and Linghe Kong. 2025. Revisiting Siamese-Based 3D Single Object Tracking With a Versatile Transformer. *IEEE Transactions on Pattern Analysis and Machine Intelligence* (2025).

[120] Kai Liu, Zhihang Fu, Sheng Jin, Ze Chen, Fan Zhou, Rongxin Jiang, Yaowu Chen, and Jieping Ye. 2024. ESOD: efficient small object detection on high-resolution images. *IEEE Transactions on Image Processing* (2024).

[121] Li Liu, Wanli Ouyang, Xiaogang Wang, Paul Fieguth, Jie Chen, Xinwang Liu, and Matti Pietikäinen. 2020. Deep learning for generic object detection: A survey. *International journal of computer vision* 128, 2 (2020), 261–318.

[122] Weiping Liu, Jia Sun, Wanyi Li, Ting Hu, and Peng Wang. 2019. Deep learning on point clouds and its application: A survey. *Sensors* 19, 19 (2019), 4188.

[123] Yajing Liu, Shijun Zhou, Xiyao Liu, Chunhui Hao, Baojie Fan, and Jiandong Tian. 2024. Unbiased faster r-cnn for single-source domain generalized object detection. In *Proceedings of the IEEE/CVF conference on computer vision and pattern recognition*. 28838–28847.

[124] Zhe Liu, Jinghua Hou, Xinyu Wang, Xiaoqing Ye, Jingdong Wang, Hengshuang Zhao, and Xiang Bai. 2024. Lion: Linear group rnn for 3d object detection in point clouds. *Advances in Neural Information Processing Systems* 37 (2024), 13601–13626.

[27] Run Luo, Zikai Song, Lintao Ma, Jinlin Wei, Wei Yang, and Min Yang. 2024. Diffusiontrack: Diffusion model for multi-object tracking. In *Proceedings of the AAAI conference on artificial intelligence*, Vol. 38. 3991–3999.

[126] Ziyang Luo, Nian Liu, Wangbo Zhao, Xuguang Yang, Dingwen Zhang, Deng-Ping Fan, Fahad Khan, and Junwei Han. 2024. Vscode: General visual salient and camouflaged object detection with 2d prompt learning. In *Proceedings of the IEEE/CVF conference on computer vision and pattern recognition*. 17169–17180.

[127] Zhipeng Luo, Changqing Zhou, Liang Pan, Gongjie Zhang, Tianrui Liu, Yueru Luo, Haiyu Zhao, Ziwei Liu, and Shijian Lu. 2024. Exploring point-BEV fusion for 3D point cloud object tracking with transformer. *IEEE transactions on pattern analysis and machine intelligence* 46, 9 (2024), 5921–5935.

[128] Weiyi Lv, Yuhang Huang, Ning Zhang, Ruei-Sung Lin, Mei Han, and Dan Zeng. 2024. Diffmot: A real-time diffusion-based multiple object tracker with nonlinear prediction. In *Proceedings of the IEEE/CVF conference on computer vision and pattern recognition*. 19321–19330.

[129] Navid Mahdian, Mohammad Jani, Amir M Soufi Enayati, and Homayoun Najjaran. 2024. Ego-motion aware target prediction module for robust multi-object tracking. *arXiv preprint arXiv:2404.03110* (2024).

[130] Muhammad Nawfal Meeran, Bhanu Pratyush Mantha, et al. 2024. Sam-pm: Enhancing video camouflaged object detection using spatio-temporal attention. In *Proceedings of the IEEE/CVF Conference on Computer Vision and Pattern Recognition*. 1857–1866.

[131] Qinghao Meng, Wenguan Wang, Tianfei Zhou, Jianbing Shen, Luc Van Gool, and Dengxin Dai. 2020. Weakly supervised 3d object detection from lidar point cloud. In *European Conference on computer vision*. Springer, 515–531.

[132] Ao Mou, Yukang Lu, Jiahao He, Dingyao Min, Keren Fu, and Qijun Zhao. 2024. Salient object detection in rgb-d videos. *IEEE Transactions on Image Processing*

[133] Pha Nguyen, Ngan Le, Jackson Cothren, Alper Yilmaz, and Khoa Luu. 2024. DINTR: Tracking via diffusion-based interpolation. *arXiv preprint arXiv:2410.10053* (2024).

[134] Chinedu Innocent Nwoye, Kareem Elgohary, Anvita Srinivas, Fauzan Zaid, Joël L Lavanchy, and Nicolas Padoy. 2025. Cholectrack20: A multi-perspective tracking dataset for surgical tools. In *Proceedings of the Computer Vision and Pattern Recognition Conference*. 8942–8952.

[135] Liang Peng, Junyuan Gao, Xinran Liu, Weihong Li, Shaohua Dong, Zhipeng Zhang, Heng Fan, and Libo Zhang. 2024. Vasttrack: Vast category visual object tracking. *Advances in Neural Information Processing Systems* 37 (2024), 130797–130818.

[136] Yafei Qi, Menghao Yang, Fan Wu, Chen Wang, and Yongmin Zhang. 2025. Harmonizing Attention Fields with Knowledge Distillation for Multi-View 3D Object Detection. In *Proceedings of the Computer Vision and Pattern Recognition Conference*. 3759–3767.

[137] Rui Qian, Xin Lai, and Xirong Li. 2022. 3D object detection for autonomous driving: A survey. *Pattern Recognition* 130 (2022), 108796.

[138] Yanyuan Qiao, Chaorui Deng, and Qi Wu. 2020. Referring expression comprehension: A survey of methods and datasets. *IEEE Transactions on Multimedia* 23 (2020), 4426–4440.

[139] Zheng Qin, Le Wang, Sanping Zhou, Panpan Fu, Gang Hua, and Wei Tang. 2024. Towards generalizable multi-object tracking. In *Proceedings of the IEEE/CVF Conference on Computer Vision and Pattern Recognition*. 18995–19004.

[140] Taki Hasan Rafi, Ratul Mahjabin, Emon Ghosh, Young-Woong Ko, and Jeong-Gun Lee. 2024. Domain generalization for semantic segmentation: a survey. *Artificial Intelligence Review* 57, 9 (2024), 247.

[141] Arjun Raj, Lei Wang, and Tom Gedeon. 2025. Tracknetv4: Enhancing fast sports object tracking with motion attention maps. In *ICASSP 2025-2025 IEEE International Conference on Acoustics, Speech and Signal Processing (ICASSP)*. IEEE, 1–5.

[142] Yasiru Ranasinghe, Deepti Hegde, and Vishal M Patel. 2024. Monodiff: Monocular 3d object detection and pose estimation with diffusion models. In *Proceedings of the IEEE/CVF Conference on Computer Vision and Pattern Recognition*. 10659–10670.

[143] Atom Scott, Ikuma Uchida, Ning Ding, Rikuhei Umemoto, Rory Bunker, Ren Kobayashi, Takeshi Koyama, Masaki Onishi, Yoshinari Kameda, and Keisuke Fujii. 2024. Teamtrack: A dataset for multi-sport multi-object tracking in full-pitch videos. In *Proceedings of the IEEE/CVF conference on computer vision and pattern recognition*. 3357–3366.

[144] Wenxu Shi and Bochuan Zheng. 2024. Conflict-alleviated gradient descent for adaptive object detection. In *Proceedings of the Thirty-Third International Joint Conference on Artificial Intelligence*. 1236–1244.

[145] Kyujin Shim, Kangwook Ko, Yujin Yang, and Changick Kim. 2025. Focusing on Tracks for Online Multi-Object Tracking. In *Proceedings of the Computer Vision and Pattern Recognition Conference*. 11687–11696.

[146] Kevin Smith, Daniel Gatica-Perez, J Odobez, and Sileye Ba. 2005. Evaluating multi-object tracking. In *2005 IEEE computer society conference on computer vision and pattern recognition (CVPR'05)-workshops*. IEEE, 36–36.

[147] Xiuqiang Song, Li Jin, Zhengxian Zhang, Jiachen Li, Fan Zhong, Guofeng Zhang, and Xueying Qin. 2025. Prior-free 3D Object Tracking. In *Proceedings of the Computer Vision and Pattern Recognition Conference*. 1200–1209.

[148] Yan Song, Zheng Hu, Tiancheng Li, and Hongqi Fan. 2022. Performance evaluation metrics and approaches for target tracking: A survey. *Sensors* 22, 3 (2022), 793.

[149] Zikai Song, Run Luo, Lintao Ma, Ying Tang, Yi-Ping Phoebe Chen, Junqing Yu, and Wei Yang. 2025. Temporal Coherent Object Flow for Multi-Object Tracking. In *Proceedings of the AAAI Conference on Artificial Intelligence*, Vol. 39. 6978–6986.

[150] Tomasz Stanczyk, Seongro Yoon, and Francois Bremond. 2025. No Train Yet Gain: Towards Generic Multi-Object Tracking in Sports and Beyond. In *Proceedings of the Computer Vision and Pattern Recognition Conference*. 6039–6048.

[151] Jingtao Sun, Yaonan Wang, Mingtao Feng, Yulan Guo, Ajmal Mian, and Mike Zheng Shou. 2024. L4D-Track: Language-to-4D Modeling Towards 6-DoF Tracking and Shape Reconstruction in 3D Point Cloud Stream. In *Proceedings of the IEEE/CVF Conference on Computer Vision and Pattern Recognition*. 21146–21156.

[152] Yiming Sun, Fan Yu, Shaoxiang Chen, Yu Zhang, Junwei Huang, Yang Li, Chenhui Li, and Changbo Wang. 2024. Chattracker: Enhancing visual tracking performance via chatting with multimodal large language model. *Advances in Neural Information Processing Systems* 37 (2024), 39303–39324.

[153] Yuedong Tan, Jiawei Shao, Eduard Zamfir, Ruanjun Li, Zhaochong An, Chao Ma, Danda Paudel, Luc Van Gool, Radu Timofte, and Zongwei Wu. 2025. What You Have is What You Track: Adaptive and Robust Multimodal Tracking. In *Proceedings of the IEEE/CVF International Conference on Computer Vision*. 3455–3465.

[154] Torben Teepe, Philipp Wolters, Johannes Gilg, Fabian Herzog, and Gerhard Rigoll. 2024. Lifting multi-view detection and tracking to the bird's eye view. In

*Proceedings of the IEEE/CVF Conference on Computer Vision and Pattern Recognition*. 667–676.

[155] Antonio Torralba and Alexei A Efros. 2011. Unbiased look at dataset bias. In *CVPR 2011*. IEEE, 1521–1528.

[156] Dai Quoc Tran, Armstrong Aboah, Yuntae Jeon, Maged Shoman, Minsoo Park, and Seunghee Park. 2024. Low-light image enhancement framework for improved object detection in fisheye lens datasets. In *Proceedings of the IEEE/CVF Conference on Computer Vision and Pattern Recognition*. 7056–7065.

[157] Chen Wang, Liyuan Zhang, Le Hui, Qi Liu, and Yuchao Dai. 2025. Geometry-Aware 3D Salient Object Detection Network. In *Proceedings of the AAAI Conference on Artificial Intelligence*, Vol. 39. 7554–7562.

[158] Gaoang Wang, Mingli Song, and Jenq-Neng Hwang. 2022. Recent advances in embedding methods for multi-object tracking: A survey. *arXiv preprint arXiv:2205.10766* (2022).

[159] Junke Wang, Zuxuan Wu, Dongdong Chen, Chong Luo, Xiyang Dai, Lu Yuan, and Yu-Gang Jiang. 2025. Omnitracker: Unifying visual object tracking by tracking-with-detection. *IEEE Transactions on Pattern Analysis and Machine Intelligence* (2025).

[160] Kunpeng Wang, Keke Chen, Chenglong Li, Zhengzheng Tu, and Bin Luo. 2025. Alignment-Free RGB-T Salient Object Detection: A Large-scale Dataset and Progressive Correlation Network. In *Proceedings of the AAAI Conference on Artificial Intelligence*, Vol. 39. 7780–7788.

[161] Liqiong Wang, Jinyu Yang, Yanfu Zhang, Fangyi Wang, and Feng Zheng. 2024. Depth-aware concealed crop detection in dense agricultural scenes. In *Proceedings of the IEEE/CVF conference on computer vision and pattern recognition*. 17201–17211.

[162] Qitai Wang, Yuntao Chen, and Zhaoxiang Zhang. 2025. Uncertain Object Representation for Image-Based 3D Object Perception. *IEEE Transactions on Pattern Analysis and Machine Intelligence* (2025).

[163] Xiao Wang, Yu Jin, Wentao Wu, Wei Zhang, Lin Zhu, Bo Jiang, and Yonghong Tian. 2025. Object detection using event camera: A moe heat conduction based detector and a new benchmark dataset. In *Proceedings of the Computer Vision and Pattern Recognition Conference*. 29321–29330.

[164] Xinzhe Wang, Kang Ma, Qiankun Liu, Yunhao Zou, and Ying Fu. 2024. Multi-object tracking in the dark. In *Proceedings of the IEEE/CVF Conference on Computer Vision and Pattern Recognition*. 382–392.

[165] Xiao Wang, Shiao Wang, Chuanming Tang, Lin Zhu, Bo Jiang, Yonghong Tian, and Jin Tang. 2024. Event stream-based visual object tracking: A high-resolution benchmark dataset and a novel baseline. In *Proceedings of the IEEE/CVF Conference on Computer Vision and Pattern Recognition*. 19248–19257.

[166] Yanjie Wang, Xu Zou, Luxin Yan, Sheng Zhong, and Jiahuan Zhou. 2024. Snida: Unlocking few-shot object detection with non-linear semantic decoupling augmentation. In *Proceedings of the IEEE/CVF Conference on Computer Vision and Pattern Recognition*. 12544–12553.

[167] Yu-Hsiang Wang, Jun-Wei Hsieh, Ping-Yang Chen, Ming-Ching Chang, Hung-Hin So, and Xin Li. 2024. Smiletrack: Similarity learning for occlusion-aware multiple object tracking. In *Proceedings of the AAAI conference on artificial intelligence*, Vol. 38. 5740–5748.

[168] Zhen Wang, Dongyuan Li, Yaozu Wu, Tianyu He, Jiang Bian, and Renhe Jiang. 2024. Diffusion models in 3d vision: A survey. *arXiv preprint arXiv:2410.04738* (2024).

[169] Hongkai Wei, Yang Yang, Shijie Sun, Mingtao Feng, Xiangyu Song, Qi Lei, Hongli Hu, Rong Wang, Huansheng Song, Naveed Akhtar, et al. 2025. Mono3DVLT: Monocular-Video-Based 3D Visual Language Tracking. In *Proceedings of the Computer Vision and Pattern Recognition Conference*. 13886–13896.

[170] Ranwan Wu, Tian-Zhu Xiang, Guo-Sen Xie, Rongrong Gao, Xiangbo Shu, Fang Zhao, and Ling Shao. 2025. Uncertainty-aware transformer for referring camouflaged object detection. *IEEE Transactions on Image Processing* (2025).

[43] Zongwei Wu, Jilai Zheng, Xiangxuan Ren, Florin-Alexandru Vasluianu, Chao Ma, Danda Pani Paudel, Luc Van Gool, and Radu Timofte. 2024. Single-model and any-modality for video object tracking. In *Proceedings of the IEEE/CVF conference on computer vision and pattern recognition*. 19156–19166.

[172] Xing Xi, Yangyang Huang, Jinhao Lin, and Ronghua Luo. 2024. Ktcn: Enhancing open-world object detection with knowledge transfer and class-awareness neutralization. In *Proceedings of the Thirty-Third International Joint Conference on Artificial Intelligence, IJCAI-24*. 1462–1470.

[173] Chao Xiao, Wei An, Yifan Zhang, Zhuo Su, Miao Li, Weidong Sheng, Matti Pietikäinen, and Li Liu. 2024. Highly efficient and unsupervised framework for moving object detection in satellite videos. *IEEE Transactions on Pattern Analysis and Machine Intelligence* 46, 12 (2024), 11532–11539.

[44] Fei Xie, Zhongdao Wang, and Chao Ma. 2024. Diffusiontrack: Point set diffusion model for visual object tracking. In *Proceedings of the IEEE/CVF Conference on Computer Vision and Pattern Recognition*. 19113–19124.

[175] Timing Yang, Yuanliang Ju, and Li Yi. 2024. ImOV3D: Learning Open Vocabulary Point Clouds 3D Object Detection from Only 2D Images. *Advances in Neural Information Processing Systems* 37 (2024), 141261–141291.

[176] Mufeng Yao, Jinlong Peng, Qingdong He, Bo Peng, Hao Chen, Mingmin Chi, Chao Liu, and Jon Atli Benediktsson. 2025. MM-Tracker: Motion Mamba for UAV-platform Multiple Object Tracking. In *Proceedings of the AAAI Conference on Artificial Intelligence*, Vol. 39. 9409–9417.

[177] Sheng Ye, Xin Chen, Yan Zhang, Xianming Lin, and Liujuan Cao. 2025. Escnet: Edge-semantic collaborative network for camouflaged object detection. In *Proceedings of the IEEE/CVF International Conference on Computer Vision*. 20053–20063.

[178] Jiong YIN, Zhe-Dong ZHANG, Yu-Han GAO, Zhi-Wen YANG, Liang LI, Mang XIAO, Yao-Qi SUN, and Cheng-Gang YAN. 2023. Survey on Vision-language Pre-training. *Journal of Software* 34, 5 (2023), 2000–2023.

[179] Xiaotong Yu, Yang Wang, and Haijing Sun. 2025. Open-vocabulary Object Detection Survey. *Procedia Computer Science* 266 (2025), 62–71.

[180] Yi Yu, Xue Yang, Yansheng Li, Zhenjun Han, Feipeng Da, and Junchi Yan. 2025. Wholly-WOOD: Wholly Leveraging Diversified-quality Labels for Weakly-supervised Oriented Object Detection. *IEEE Transactions on Pattern Analysis and Machine Intelligence* (2025).

[181] Yuhang Zang, Wei Li, Jun Han, Kaiyang Zhou, and Chen Change Loy. 2025. Contextual object detection with multimodal large language models. *International Journal of Computer Vision* 133, 2 (2025), 825–843.

[182] Kai Zeng, Qian Ma, Jia Wen Wu, Zhe Chen, Tao Shen, and Chenggang Yan. 2022. FPGA-based accelerator for object detection: a comprehensive survey. *Journal of Supercomputing* 78, 12 (2022).

[183] Rui Zeng, Yuanzhou Huang, and Songwei Pei. 2025. TGFormer: Transformer with Track Query Group for Multi-Object Tracking. In *Proceedings of the AAAI Conference on Artificial Intelligence*, Vol. 39. 9824–9832.

[184] Dingwen Zhang, Junwei Han, Gong Cheng, and Ming-Hsuan Yang. 2021. Weakly supervised object localization and detection: A survey. *IEEE transactions on pattern analysis and machine intelligence* 44, 9 (2021), 5866–5885.

[185] Gang Zhang, Junnan Chen, Guohuan Gao, Jianmin Li, Si Liu, and Xiaolin Hu. 2024. Safdnet: A simple and effective network for fully sparse 3d object detection. In *Proceedings of the IEEE/CVF Conference on Computer Vision and Pattern Recognition*. 14477–14486.

[186] Guowen Zhang, Lue Fan, Chenhang He, Zhen Lei, Zhaoxiang Zhang, and Lei Zhang. 2024. Voxel mamba: Group-free state space models for point cloud based 3d object detection. *Advances in Neural Information Processing Systems* 37 (2024), 81489–81509.

[187] Hao Zhang, Aixin Sun, Wei Jing, and Joey Tianyi Zhou. 2023. Temporal sentence grounding in videos: A survey and future directions. *IEEE Transactions on Pattern Analysis and Machine Intelligence* 45, 8 (2023), 10443–10465.

[188] Jingyi Zhang, Jiaxing Huang, Sheng Jin, and Shijian Lu. 2024. Vision-language models for vision tasks: A survey. *IEEE transactions on pattern analysis and machine intelligence* 46, 8 (2024), 5625–5644.

[189] Jialu Zhang, Xiaoying Yang, Wentao He, Jianfeng Ren, Qian Zhang, Yitian Zhao, Ruibin Bai, Xiangjian He, and Jiang Liu. 2024. Scale optimization using evolutionary reinforcement learning for object detection on drone imagery. In *Proceedings of the AAAI Conference on Artificial Intelligence*, Vol. 38. 410–418.

[190] Peng Zhang, Xin Li, Liang He, and Xin Lin. 2023. 3d multiple object tracking on autonomous driving: A literature review. *arXiv preprint arXiv:2309.15411* (2023).

[191] Xuying Zhang, Bowen Yin, Zheng Lin, Qibin Hou, Deng-Ping Fan, and Ming-Ming Cheng. 2025. Referring camouflaged object detection. *IEEE Transactions on Pattern Analysis and Machine Intelligence* (2025).

[192] Yupeng Zhang, Ruize Han, Fangnan Zhou, Song Wang, Wei Feng, and Liang Wan. 2025. ODOV: Towards Open-Domain Open-Vocabulary Object Detection. *arXiv preprint arXiv:2508.01253* (2025).

[193] Yunhan Zhao, Haoyu Ma, Shu Kong, and Charless Fowlkes. 2024. Instance tracking in 3D scenes from egocentric videos. In *Proceedings of the IEEE/CVF Conference on Computer Vision and Pattern Recognition*. 21933–21944.

[194] Guangze Zheng, Shijie Lin, Haobo Zuo, Changhong Fu, and Jia Pan. 2024. Nettrack: Tracking highly dynamic objects with a net. In *Proceedings of the IEEE/CVF Conference on Computer Vision and Pattern Recognition*. 19145–19155.

[195] Jingchun Zhou, Zongxin He, Dehuan Zhang, Siyuan Liu, Xianping Fu, and Xuelong Li. 2025. Spatial residual for underwater object detection. *IEEE Transactions on Pattern Analysis and Machine Intelligence* (2025).

[196] Xingcheng Zhou, Mingyu Liu, Ekim Yurtsever, Bare Luka Zagar, Walter Zimmer, Hu Cao, and Alois C Knoll. 2024. Vision language models in autonomous driving: A survey and outlook. *IEEE Transactions on Intelligent Vehicles* (2024).

[197] Chaoyang Zhu and Long Chen. 2024. A survey on open-vocabulary detection and segmentation: Past, present, and future. *IEEE Transactions on Pattern Analysis and Machine Intelligence* 46, 12 (2024), 8954–8975.

[198] Hao Zhu, Man-Di Luo, Rui Wang, Ai-Hua Zheng, and Ran He. 2021. Deep audio-visual learning: A survey. *International Journal of Automation and Computing* 18, 3 (2021), 351–376.

[199] Zhengxia Zou, Keyan Chen, Zhenwei Shi, Yuhong Guo, and Jieping Ye. 2023. Object detection in 20 years: A survey. *Proc. IEEE* 111, 3 (2023), 257–276.