# OpenReview forum: "Rethinking Object Detection and Tracking"
_ACM.org/TheWebConf/2026/Workshop/TIME — TIME 2026 Oral_

### Official Review · Reviewer_raRu · 2026-01-04
**Challenge-driven survey with strong insights, small methodological gaps**

**Rating:** 7
**Confidence:** 3

**Review:**

## Originality and significance
**Strengths**

* This is a timely, substantial, and thoughtfully argued survey that synthesizes a rapidly evolving literature through a challenge-centric lens. The narrative integrates architectural, modality, and supervision trends and articulates clear, deployment-relevant insights and open problems. The added value over task-centric surveys is evident, and the proposed framing - particularly around unified temporal reasoning, open-world semantics, and uncertainty - is compelling and useful to the community.

* The challenge-centric reframing encourages evaluation and design choices aligned with deployment realities, bridging fragmented subfields and incentivizing solutions that generalize beyond legacy benchmarks

**Weaknesses**

* To reach top-tier impact, the paper should strengthen methodological transparency of its quantitative meta-analysis:  clear inclusion/exclusion criteria and how overlapping categories are handled

## Technical content and experimental validation
**Strengths**

* Reframes the field around challenge-driven axes (task formulation, supervision/openness, spatial–temporal scope, modality) rather than task labels, offering a fresh, integrative perspective

* The identified open problems (open-world, unified evaluation, uncertainty, multimodality) are well chosen and likely to guide research agendas across academia and industry

**Weaknesses**

* The survey claims 2022-2025 coverage, while Section 3’s quantitative analysis focuses on 2024-2025. What motivated this time restriction, and do the main quantitative conclusions hold if 2022-2023 papers are included?

## Clarity
**Strengths**

* The paper is organized in a clear manner - from motivation and related work through methodology (challenge-driven framework) to analysis and future directions - making it easy to follow the narrative

**Weaknesses**

* It might be helpful to add axis labels in the figure rather than explaining them only in the caption

---

### Official Review · Reviewer_6X3G · 2026-01-07
**Survey of Object Detection and Tracking: Strong Content but Non-Compliant Length**

**Rating:** 8
**Confidence:** 4

**Review:**

This paper provides a comprehensive and well-executed survey of recent research on object detection and tracking, with an explicit emphasis on challenge-driven analysis rather than task-centric categorization. The manuscript demonstrates strong technical understanding, broad coverage of the literature, and a clear effort to synthesize trends across architectures, modalities, datasets, and evaluation practices. The quantitative trend analysis and unified tables are particularly useful; the discussion of emerging paradigms and open problems is thoughtful and well-articulated.

The paper is clearly written and logically structured in terms of its content. The logical movement from taxonomy to quantitative analysis into a forward-looking discussion makes the survey easy to follow despite its scope. Overall, the framing centered around challenges does indeed provide broad insights that would go beyond the simple enumeration of methods; such work could be useful for researchers entering into the field for long-form references.

Strengths:
 - Comprehensive, up-to-date coverage of object detection and tracking literature.
- Clearly challenge-driven organization; cross-task connections indicated.
- Useful quantitative analysis of datasets, metrics, and research trends.
- Well-written and technically sound synthesis.

Overall, the paper is solid in content.

---

### Official Review · Reviewer_WkX1 · 2026-01-07
**Rethinking Object Detection and Tracking: A Challenge-Driven Synthesis This review of 100+ works identifies a paradigm shift from task-isolated pipelines toward unified, open-world perception systems that address real-world deployment challenges through multimodal fusion and generative temporal modeling**

**Rating:** 8
**Confidence:** 4

**Review:**

Based on the provided sources, here is an evaluation of the work titled **"Rethinking Object Detection and Tracking"** (2025).

### **1. Quality of the Work**
The quality of this work is exceptionally high, characterised by a **rigorous and disciplined curation process** for selecting literature from premier computer vision and machine learning venues (such as CVPR, ICCV, and NeurIPS). It goes beyond a simple summary by performing a **quantitative meta-analysis** of dataset usage, evaluation metrics, and the prevalence of specific research challenges over time.

The work is grounded in a massive review of over one hundred representative papers published between 2022 and 2025, ensuring that the findings are representative of the most current state-of-the-art. The inclusion of a detailed literature review table [151–203] allows for granular comparisons of model categories, datasets, and targeted challenges, such as **occlusion**, **fast motion**, and **domain shift**.

### **2. Clarity of Presentation**
The clarity of the survey is one of its strongest attributes. It is organised around a clear, multi-dimensional taxonomy:
*   **Task Formulation:** The shift from modular to unified pipelines.
*   **Supervision:** Scaling from dense annotations to weakly/self-supervised methods.
*   **Spatial-Temporal Scope:** Expanding from 2D frames to persistent 3D perception.
*   **Sensing Modality:** Moving from single sensors (RGB) to multimodal fusion.

The authors effectively use figures and tables to illustrate complex trends, such as the dominance of established benchmarks like $MOT17$ and $COCO$ compared to the rising use of 3D and vision-language datasets. This structured approach ensures that the "conceptual shift" in the field—from task-isolated pipelines to semantically grounded systems—is easily understood by the reader.

### **3. Originality of Approach**
This work demonstrates significant originality by adopting a **challenge-driven synthesis** rather than a traditional task-centric one. While previous surveys typically isolate detection from tracking or 2D from 3D, this paper frames the literature through fundamental bottlenecks that limit real-world deployment, such as:
*   **Long-term identity association**.
*   **Non-linear and fast motion**.
*   **Uncertainty modeling in adverse environments**.

By extracting "cross-task patterns" that are often overlooked, the authors provide a unique perspective on the convergence of different sub-fields, such as how **diffusion-based temporal modeling** is being applied across both detection and tracking tasks to handle motion uncertainty.

### **4. Significance of the Research**
The significance of this work lies in its role as a **diagnostic tool and roadmap** for the future of visual perception. It identifies a critical mismatch between current benchmark-driven optimization and the requirements of real-world deployment.

Specifically, the paper highlights:
*   The **gradual erosion of rigid task boundaries**, favouring unified formulations that reason about localization and identity jointly.
*   The emergence of **open-vocabulary** and **language-conditioned tracking**, which expands perceptual flexibility but introduces new stability challenges.
*   The **pressing need for unified evaluation protocols** that measure temporal consistency and uncertainty rather than just spatial accuracy.

---

### **Pros and Cons of the Work**

**Pros:**
*   **Comprehensive Scope:** Covers a wide range of modalities (RGB, LiDAR, Thermal, Event-based) and tasks (2D/3D MOT, SOT, point tracking).
*   **Forward-Looking:** Identifies and analyzes emerging trends like **Mamba-based state-space models** and **diffusion models** for trajectory prediction.
*   **Quantitative Insights:** Provides empirical data on which challenges (e.g., $occlusion$) and metrics (e.g., $mAP$, $HOTA$, $IDF1$) dominate the current research landscape.
*   **Actionable Guidance:** Offers practical advice on when to use specific architectures, such as choosing Transformers for long-term interactions versus state-space models for real-time constraints.
*   **Emphasis on Real-World Deployment:** Explicitly addresses bottlenecks like data scarcity, annotation efficiency, and robustness in safety-critical applications.

**Cons:**
*   **Benchmark Concentration Risk:** The analysis reveals that research is still heavily biased toward a small set of "idealized" benchmarks, which may limit the generalizability of the surveyed methods to true open-world scenarios.
*   **High Architectural Complexity:** While the paper notes the rise of unified models, it acknowledges that these often come at the cost of increased computational overhead and data hunger.
*   **Fragmentation of Evaluation:** The work highlights that the proliferation of task-specific metrics makes cross-method comparison increasingly difficult, a problem for which a universal solution is still being sought.
*   **Inference Latency in Generative Models:** It notes that diffusion-based trackers, while robust, currently face limitations in inference speed, potentially hindering their use in real-time systems.

### **Metaphor for Understanding**
To understand the conceptual shift described in the sources, imagine the field of computer vision as a **construction project**.

Previously, researchers worked in **isolated silos**: one team built the windows (detection), while another built the hallways (tracking). This survey suggests that the field is now moving toward **modular, unified architecture**. Instead of building windows and hallways separately, they are creating a single, integrated "smart frame" (unified temporal reasoning) that can adapt its shape based on the weather (multimodal fusion) and instructions from the owner (language grounding), ensuring the entire building remains stable even during an earthquake (occlusion or fast motion).

---

### Official Review · Reviewer_Y6bd · 2026-01-07
**Percipient Survey in Object Detection and Tracking**

**Rating:** 7
**Confidence:** 5

**Review:**

This survey in object detection and tracking provided literature review in model architectures, datasets, evaluation protocols, and open issues.

Authors understand the research trends so that they lifted research issues to upper levels and made significantly contributions in summarizing taxonomy of detection and tracking paradigms, quantitative trend analysis, analysis and discussion, open challenges and future directions. Also, discussions in each aspect were categorized very well.

The authors can add citations to following discussions to make the summary more trackable. Evaluation metrics can be discussed a little more on strength and weakness. Figure 1 and 2 can be formatted better if the x-axes is labeled.

Taxonomy of detection and tracking paradigms
1. task formulation: from modular pipelines to unified temporal reasoning
2. supervision and openness: scaling beyond exhaustive annotations.
3. spatial and temporal scope: from 2d frames to persistent 3d perception
4. sensing modality: from single sensors to multimodal perception

quantitative trend analysis
1. analysis protocol and scope
2. task formulation trends and convergence
3. architectural evolution
4. modalities and sensor diversity
5. targeted challenges and research gaps
6. temporal dynamics and emerging directions
7. datasets and evaluation practices


analysis and discussion
1. challenge-driven convergence of detection and tracking
2. architectural trade-offs and practical guidance
3. detection paradigms and their impact on temporal consistency
4. multimodal and 3D perception as a unifying trend
5. evaluation practices, biases, and deployment implications.


open challenges and future directions
1. scalable open-world detection and tracking
2. unified evaluation protocols across tasks and modalities
3. advancing long-term temporal reasoning
4. efficient integration of large multimodal models
5. annotation-efficient and self-supervised learning
6. robustness, uncertainty, and reliability
7. towards holistic perception systems

---

### Author Rebuttal · Authors · 2026-01-12

We sincerely thank the reviewers and Program Chairs for their careful reading of our manuscript, their constructive feedback, and their positive assessment of the contribution, clarity, and significance of our work. We are particularly grateful for the recognition of the paper’s challenge-driven framing, comprehensive coverage, and quantitative synthesis. We believe that the suggestions provided have substantially helped improve the clarity, transparency, and overall quality of the manuscript. Below, we respond to each comment point by point and describe the corresponding revisions.

Response to Reviewers and Program Chairs

Response to Reviewer Y6bd07

**Comment:** The authors can add citations to the taxonomy and discussion sections to make the summary more trackable.

**Response:** We thank the reviewer for this helpful suggestion. We have added explicit citations throughout the taxonomy and discussion sections, particularly in the subsections on task formulation, supervision and openness, spatial--temporal scope, and sensing modalities. We hope that these additions improve traceability and better ground our conceptual claims in representative prior work.

**Comment:** Evaluation metrics can be discussed a little more in terms of their strengths and weaknesses.

**Response:** We appreciate this valuable suggestion. In response, we have expanded the discussion of evaluation metrics in the main text and added a dedicated appendix section that provides a glossary of commonly used metrics, including their full names, intended use cases, and what they emphasize (e.g., spatial accuracy vs.\ identity consistency). We hope that this addition clarifies both the strengths and limitations of prevalent metrics and makes the survey more self-contained.



**Comment:** Figure 1 and Figure 2 can be formatted better if the x-axes are labeled.

**Response:** We agree with this observation. We have revised all figures to include explicit x- and y-axis labels. In addition, we clarified the use of logarithmic scales directly in the captions, which we hope improves interpretability and reduces ambiguity.

Response to Reviewer WkX107

**Comment:** The reviewer highlights the conceptual clarity and originality of the work, and emphasizes the importance of grounding high-level claims in empirical evidence.

**Response:** We sincerely thank the reviewer for this encouraging assessment. To further strengthen the empirical grounding of our claims, we have improved the methodological transparency of the quantitative analysis section. Specifically, we now explicitly describe the inclusion and exclusion criteria, clarify how categories are annotated, and explain how overlapping categories are handled using a multi-label scheme. We hope these additions make the reported trends more reproducible and easier to interpret.

Response to Reviewer 6X3G07

**Comment:** The paper would benefit from improved methodological clarity.

**Response:** We appreciate this important point. In response, we have added a dedicated methodological clarification subsection at the beginning of the quantitative analysis section. This subsection now explicitly states our paper selection criteria, excluded categories of work, and the rationale for our annotation and grouping strategy. We hope this improves the transparency and reproducibility of the meta-analysis.

Response to Reviewer raRu05

**Comment:** The paper should strengthen methodological transparency of its quantitative meta-analysis: clear inclusion/exclusion criteria and how overlapping categories are handled.

**Response:** We fully agree with this concern. We have added a new paragraph that explicitly states our inclusion and exclusion criteria and clarifies our use of a multi-label annotation scheme. This explains how a single paper may contribute to multiple categories (e.g., modality, task, challenge), reflecting the inherently overlapping nature of modern research directions.



**Comment:** The survey claims 2022--2025 coverage, while Section 3 focuses on 2024--2025. What motivated this restriction?

**Response:** We thank the reviewer for raising this point. We have added a clear justification at the beginning of the quantitative analysis section. While the survey conceptually covers 2022--2025, we intentionally restrict the quantitative trend analysis to 2024--2025 in order to highlight the most recent paradigm shifts and emerging practices. Earlier works (2022--2023) are extensively incorporated into the related work, taxonomy, and qualitative synthesis sections, where they provide foundational context. We hope this distinction is now clearer.



**Comment:** It might be helpful to add axis labels in the figure rather than explaining them only in the caption.

**Response:** We agree. We have added explicit axis labels to all figures and ensured that scale choices (e.g., logarithmic axes) are clearly marked both visually and in the captions.

Response to Program Chair Comments

**Comment:** Fig.~2 seems unclear, and the coverage appears limited to 2024--2025 despite claiming 2022--2025.

**Response:** We appreciate this observation. We clarified this distinction explicitly in the text. The conceptual scope of the survey spans 2022--2025, while the quantitative analysis focuses on 2024--2025 to capture the most recent shifts. This design choice is now clearly stated and justified.


**Comment:** Why and how were the methods in the appendix selected?

**Response:** We have added a paragraph describing the selection process, including explicit inclusion and exclusion criteria, to clarify how the appendix tables were curated. We also explicitly state that our focus is on representative, peer-reviewed works that introduce, benchmark, or substantially analyze detection and tracking paradigms.



**Comment:** Most metrics are not properly explained.

**Response:** We agree with this concern and have added a dedicated appendix section containing a glossary of evaluation metrics, including their full names, what each metric measures, and its intended use. We hope this makes the paper more accessible and self-contained.



**Comment:** There is a lack of description of how challenges, datasets, and metrics were obtained.

**Response:** We have added a detailed description of the manual extraction and annotation process, clarifying how information was collected and categorized from each paper.



**Comment:** Fig. 3 horizontal axis seems incorrect.

**Response:** We have corrected the axis labeling and verified that all scales and tick marks are now consistent with the underlying data.



**Comment:** The appendix table is too long; a summary table should be added.

**Response:** We have added a compact, high-level summary table in the main paper that distills the dominant trends in task formulation, model paradigms, sensing modalities, evaluation practices, and challenges. We hope this improves readability and complements the detailed tables in the appendix.

We sincerely thank the reviewers and Program Chairs for their constructive feedback. We believe the manuscript is substantially stronger, clearer, and more transparent as a result of these suggestions, and we hope that the revised version meets the expectations for a high-quality, publicly accessible survey.

---

### Meta-Review · Area_Chair_RwVg · 2026-01-16

**Recommendation:** Accept (Oral)
**Confidence:** 5

**Metareview:**

This paper presents a comprehensive, challenge-driven survey of object detection and tracking research from recent years, synthesizing over 100 works to reveal a paradigm shift toward unified, open-world, and multimodal temporal perception systems.

Reviewers consistently praised the paper’s originality, clarity, and significance, highlighting its challenge-centric framing, comprehensive coverage, quantitative meta-analysis, and value as a roadmap for future research. The main suggestions focused on improving methodological transparency, clarifying the temporal scope of the quantitative analysis, adding clearer figure axis labels, and providing more explicit explanations of evaluation metrics, datasets, and selection criteria.

The paper’s strengths lie in its integrative taxonomy, well-structured synthesis, and insightful discussion of emerging trends and open challenges. Its weaknesses were primarily related to presentation and methodological clarity rather than technical substance.

The authors’ rebuttal thoroughly addressed all major concerns by adding clarifications, justifications, revised figures, and dedicated appendix material, substantially strengthening transparency and readability.

Based on the reviews and rebuttal, the recommendation is to accept this paper.

---

### Decision · Program_Chairs · 2026-01-16

Accept (Oral)